

# Pan-Arctic Sea Ice Concentration from SAR and Passive Microwave

Tore Wulf[1], Jørgen Buus-Hinkler[1], Suman Singha[1], Hoyeon Shi[1], and Matilde Brandt Kreiner[1]

[1]National Center for Climate Research, Danish Meteorological Institute, Copenhagen, Denmark

**Correspondence:** Tore Wulf (twu@dmi.dk)

**Abstract.** Arctic sea ice monitoring is a fundamental prerequisite for anticipating and mitigating the impacts of climate change. Satellite-based sea ice observations have been subject to intense attention over the last few decades, with passive microwave (PMW) radiometers being the primary sensors for retrieving pan-Arctic sea ice concentration, albeit with coarse spatial resolutions of a few or even tens of kilometers. Space-borne Synthetic Aperture Radar (SAR) missions, such as Sentinel-1,

provide dual-polarized C-band images with <100 meter spatial resolution, which are particularly well-suited for retrieving high-resolution sea ice information. In recent years, deep learning-based vision methodologies have emerged with promising results for SAR-based sea ice concentration retrievals. Despite recent advancements, most contributions focus on regional or local applications without empirical studies on the generalization of the algorithms to the pan-Arctic region. Furthermore, many contributions omit uncertainty quantification from the retrieval methodologies, which is a prerequisite for the integration

of automated SAR-based sea ice products into the workflows of the national ice services, or for the assimilation into numerical ocean-sea-ice coupled forecast models. Here, we present ASIP (Automated Sea Ice Products): a new and comprehensive deep learning-based methodology to retrieve high-resolution sea ice concentration with accompanying well-calibrated uncertainties from Sentinel-1 SAR and Advanced Microwave Scanning Radiometer 2 (AMSR2) passive microwave observations at a pan-Arctic scale for all seasons. We compiled a vast matched dataset of Sentinel-1 HH/HV imagery and AMSR2 brightness

temperatures to train ASIP with regional ice charts as labels. ASIP achieves an $R^2$-score of 95% against a held-out test dataset of regional ice charts. In a comparative study against pan-Arctic ice charts and PMW-based sea ice products, we show that ASIP generalizes well to the pan-Arctic region. Additionally, the comparison reveals that ASIP consistently produces relatively higher sea ice concentration than the PMW-based sea ice product, with mean biases ranging from 1.45% to 8.55%, and that the discrepancies are primarily attributed to disparities in the marginal ice zone.

## 1   Introduction

The Arctic region, characterized by its extreme climate and dynamic environmental conditions, plays a pivotal role in the Earth's climate and ecosystem (Moon et al., 2023). Among the most significant indicators of its changing state are the sea ice extent and thickness, whose dynamics are integral to understanding the broader implications of climate change (Forster et al., 2021). Monitoring sea ice parameters across the vast expanse of the Arctic is essential for tracking these changes and assessing

their impacts.



The current state-of-the-art in pan-Arctic sea ice concentration (SIC) retrieval primarily relies on passive microwave (PMW) sensors (Cavalieri et al., 1984; Andersen et al., 2007; Tonboe et al., 2016; Lavergne et al., 2019), which provide global coverage but suffer from relatively coarse spatial resolution (Feng et al., 2023). PMW-based products are crucial for monitoring long-term trends, and while some experimental products offer grid resolutions as high as 3.125 km (Meier and Stewart, 2020), they often struggle to capture fine-scale features and changes in the sea ice. Consequently, there is a pressing need for high-resolution, all-weather monitoring techniques that can offer more detailed insights into Arctic sea ice dynamics. This includes an increasing demand for near real-time high-resolution sea ice products suitable for tactical navigation from a growing maritime user group accessing wider parts of the Arctic due to the retreat and thinning of the sea ice. Satellite-based SAR has emerged as a powerful tool for monitoring Arctic sea ice due to its all-weather capability and high spatial resolution ($<100\ m$). A fully automated SAR-based sea ice retrieval system has the potential to serve Arctic maritime sectors and local community needs for timely and high-resolution sea ice information in coastal as well as off-shore regions in the Arctic. Such a system can be integrated into the workflows of the national ice services to deliver valuable assistance in their daily service to maritime users. Ultimately, SAR-based sea ice retrievals can be assimilated in numerical ocean and sea ice models, improving the quality and spatial resolution of sea ice forecasts crucial for Arctic stakeholders (Ponsoni et al., 2023).

The launch of the Sentinel-1 satellites, with their systematic and frequent acquisitions over the Arctic region, has opened up new possibilities for SAR-based SIC retrievals. While the use of traditional machine learning (ML) algorithms for SAR-based sea ice retrievals has been studied for several decades (Karvonen, 2004; Zakhvatkina et al., 2013; Ressel et al., 2016; Singha et al., 2018), the majority of recent contributions employ various modern deep learning (DL) techniques, most often convolutional neural networks (ConvNets), with promising results (Wang et al., 2016; Wulf et al., 2022; Kortum et al., 2022; Stokholm et al., 2022; Boulze et al., 2020; Malmgren-Hansen et al., 2021; Kortum et al., 2023). Despite recent strides in the predictive performance of these algorithms, most contributions focus on regional or local applications, without providing empirical studies on the generalization of the algorithms to the pan-Arctic region, thus failing to address the prospect of operational pan-Arctic sea ice products from SAR. Furthermore, only a few of the recent contributions include uncertainty quantification in the retrieval methodologies (Asadi et al., 2021; Pires de Lima and Karimzadeh, 2023; Chen et al., 2023), which is a prerequisite for the integration of automated SAR-based sea ice products into the workflows of the national ice services, or for the assimilation into numerical ocean-sea-ice coupled forecast models. While uncertainty estimates can be readily derived from the confidence scores that are produced by modern classification networks, it is empirically known that the confidence scores tend to be poorly calibrated (Guo et al., 2017; Lakshminarayanan et al., 2017; Thulasidasan et al., 2020; Ovadia et al., 2019). In other words, the confidence scores provided by modern classification networks do not accurately reflect their predictive uncertainties. Therefore, in order to derive meaningful uncertainties, the confidence scores first need to be *re*-calibrated (Guo et al., 2017; Müller et al., 2020; Lakshminarayanan et al., 2017).

As the training of deep learning models normally requires large amounts of training data, the success of these algorithms applied to SAR-based sea ice mapping can in part be attributed to the large volumes of freely available observational datasets from initiatives such as EU's Copernicus Programme. In the case of supervised deep learning, however, the process of collecting or generating the necessary label data to accompany the satellite observations can be cumbersome. This is a persisting



challenge in the Arctic domain, particularly, due to the harsh - and sometimes inaccessible - environment as well as the dynamic nature of drifting sea ice. In-situ measurements are an attractive, but scarce option for label data. In-situ measurements are generally accurate, but suffer from limited spatial and temporal coverage, not fully capturing the location-dependent seasonal variation of the state of the Arctic sea ice. Some studies create their own labels by manually delineating sea ice in the

SAR imagery, e.g. the work of Kortum et al. (2022). While this approach ensures a perfect temporal match between the SAR image and label, thus avoiding any potential sea ice drift between the SAR image and the label, the process of manual label creation is time-consuming and resource-intensive. Further, this process can be error-prone due to inherent ambiguities in the SAR imagery that might require extensive experience and expert knowledge to resolve. A third option is to use operationally provided regional ice charts as label data. Regional ice charts are produced manually by experienced ice analysts at the national

ice services, such as the Greenland and Canadian ice services. The primary advantages of ice charts are their abundance and widespread availability, with year-round coverage of vast geographical areas and a diverse variety of sea ice conditions. A comprehensive and diverse training dataset, with rare sea ice conditions represented, is crucial for the training of robust deep learning models that generalize well beyond the geographical and temporal boundaries of the training dataset, and thus are suitable for operational use. The ice charts are often produced on the basis of Sentinel-1 SAR images, enabling very timely - if

not exact - match-ups between the ice chart and a Sentinel-1 image, which is important due to the high spatial resolution of the SAR sensor and the rapidly changing sea ice conditions. However, as the ice charts are based on the analyst's interpretation of satellite observations, there are bound to be inherent uncertainties in the ice charts, stemming from the subjectivity introduced in the manual ice charting process. Furthermore, ice charts are provided in a vector format, with polygons of relatively homogeneous sea ice conditions (on Sea Ice SIGRID-3, 2014). While the format ensures ease of use for nautical navigation, this

simplified representation of the state of the sea ice misses small-scale heterogeneity and important features that are otherwise visible in the SAR imagery, such as leads, ridges, and melt ponds.

In this paper, we present a new and comprehensive deep learning-based SIC retrieval methodology denoted *ASIP* (Automated Sea Ice Products). ASIP is an ensemble of ConvNets retrieving high-resolution SIC with accompanying well-calibrated uncertainties from Sentinel-1 SAR imagery and AMSR2 brightness temperatures. ASIP is trained on a new, vast training

dataset with Sentinel-1 HH/HV imagery and Advanced Microwave Scanning Radiometer 2 (AMSR2) brightness temperatures as input and manually produced ice charts from the Greenland and Canadian Ice Services (CIS) as labels. We explore several recalibration strategies and introduce a new metric to quantify miscalibration for imbalanced multi-class classification tasks. Using reliability diagrams, we show that our proposed metric surpasses the popular ECE (Expected Calibration Error) metric, particularly when it comes to identifying class-wise miscalibration and miscalibration across confidence regions. We propose

a new retrieval methodology to retrieve SIC and the associated uncertainty from the calibrated ensemble output. Finally, we show that ASIP generalizes well to the pan-Arctic region in all seasons in a comparative study against a well-established and operational PMW-based SIC product.



## 2 Datasets

### 2.1 ASIP/AI4Arctic Sea Ice Dataset Version 2+

For the training, calibration, and initial evaluation of our proposed SAR-based SIC retrieval, we use an extended version of the ASIP/AI4Arctic Sea Ice Dataset version 2 (ASIDv2) (Saldo et al., 2020) produced by the Danish Meteorological Institute (DMI), the Technical University of Denmark (DTU) and the Nansen Environmental and Remote Sensing Center (NERSC). ASIDv2 consists of 461 samples from 2018-2019 of Sentinel-1 imagery collocated with PMW observations from the AMSR2 instrument aboard JAXA's GCOM-W1 satellite and ice charts produced manually at the DMI Greenland Ice Service. We

geographically extend ASIDv2 by including ice charts produced manually by CIS to cover the Canadian Arctic. We also temporally extend ASIDv2 by including data from 2018 up to and including 2021. From here on, we will refer to our extended version of ASIDv2 as ASIDv2+. In 2022, we - together with partners of the AI4Arctic project - released a subset of the ASIDv2+ dataset, the AI4Arctic Sea Ice Challenge Dataset (Buus-Hinkler et al., 2022), as part of the *AutoICE* challenge, sponsored by the European Space Agency.

ASIDv2+ contains a total of 5382 samples. The geographical distribution of the samples in ASIDv2+ is shown in Figure 1. As one of the main responsibilities of the ice services is the support of shipping and maritime activities, the availability of ice charts is not equally distributed geographically, nor temporally, with most ice charts being concentrated in regions and seasons of high maritime traffic.

Figure 2 shows an example scene from the ASIDv2+ dataset off of the central-western coast of Greenland (see yellow

outline in Figure 1) from May 16th, 2021. The figure illustrates the primary contents of the dataset; Sentinel-1 HH/HV imagery, AMSR2 brightness temperatures, and SIC from a manually produced ice chart.

### 2.1.1 Sentinel-1 imagery

The Sentinel-1 L1 Ground Range Detected (GRD) products contained in the ASIDv2+ dataset were acquired in the Extra Wide (EW) and Interferometric Wide (IW) swath modes in HH/HV dual-polarisation. The Sentinel-1 imagery was denoised using

the noise vectors provided in the Sentinel-1 product metadata (Matthieu Bourbigot, 2023). Sentinel-1 EW imagery covers an area of up to roughly 400x400 $km$ with a native pixel spacing of 40 $m$, whereas Sentinel-1 IW imagery covers an area of up to 250x250 $km$ with a native pixel spacing of $10m$. The spatial resolution of the sensor is  87x93 $m$ in the range and azimuth directions, respectively, when operated in the EW mode, and  20x22 $m$ when operated in the IW mode (European Space Agency, 2023).

The SAR imagery is considered the primary source of information in our proposed sea ice retrieval. In order to preserve the detailed radiometric information in the SAR imagery, the ice chart information and AMSR2 brightness temperatures have been resampled to the SAR geometry during the generation of ASIDv2+.





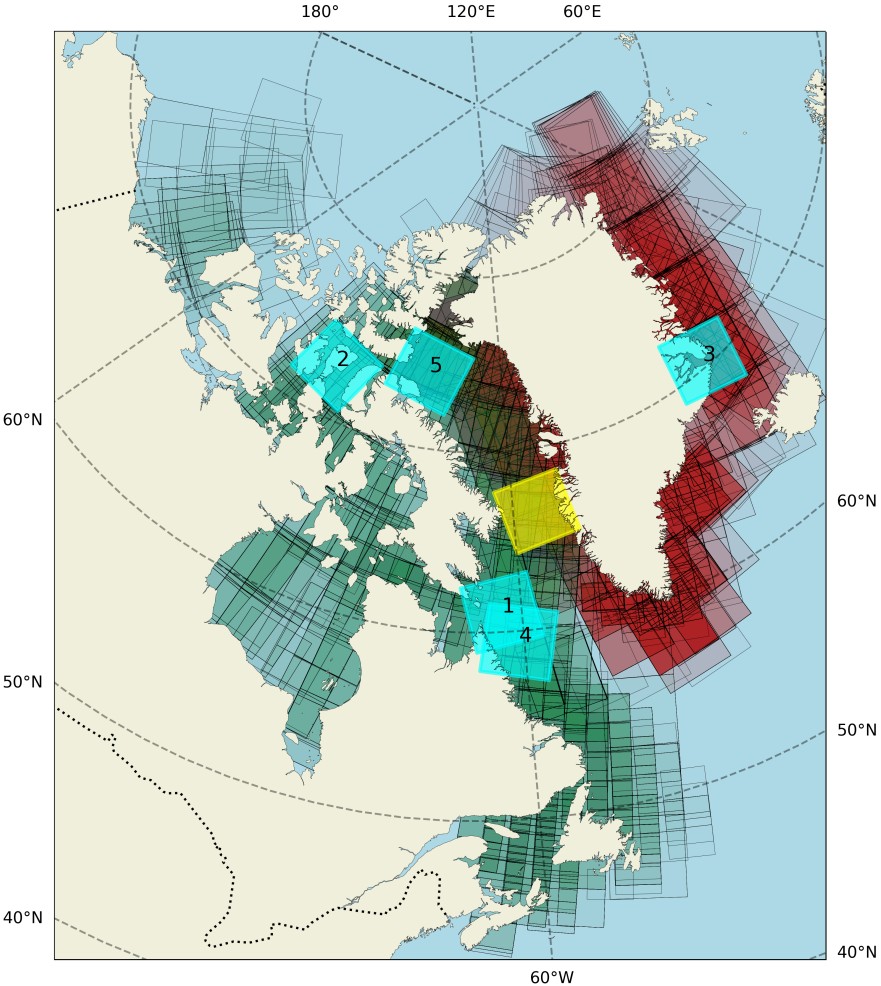

**Figure 1.** Density plot showing the geographical distribution of the 5382 scenes in the ASIDv2+ dataset used in this study. Red boxes outline Sentinel-1 scenes that have been matched up with an ice chart from the Greenland Ice Service at DMI (2978 scenes), while green squares outline Sentinel-1 scenes that have been matched up with an ice chart from the Canadian Ice Service (2404 scenes). The yellow box outlines the geographical extent of the dataset example illustrated in Figure 2. The cyan boxes 1-5 outline the geographical extents of the five example scenes shown in Figure 6.

### 2.1.2 Regional ice charts

The regional ice charts contained in ASIDv2+ are not publicly available, but were kindly provided by the Greenland Ice Service at DMI and CIS. The ice charts are drawn by experienced ice analysts in a polygonized vector format on the basis of manual interpretation of satellite observations, primarily C-band SAR images (e.g. Sentinel-1 or the Radarsat Constellation Mission (RCM)), but also auxiliary satellite observations, e.g. optical or infrared imagery, when available and advantageous.



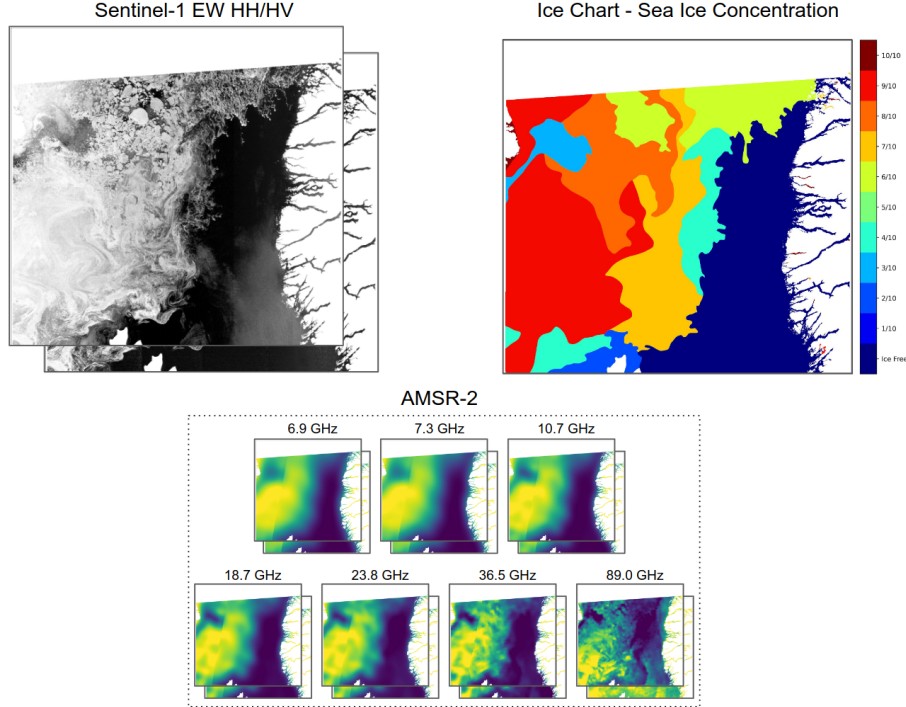

**Figure 2.** Example scene from the ASIDv2+ dataset off of the central-western coast of Greenland from May 16th, 2021. **Top left**: Sentinel-1 EW HH/HV imagery. **Top right**: Sea ice concentration from an ice chart produced by the Greenland Ice Service at DMI. **Bottom**: AMSR2 brightness temperatures.

Regional ice charts produced by DMI and CIS follow the World Meteorological Organization's (WMO) sea ice nomenclature and the charts are provided in the SIGRID3 vector format for archiving digital ice charts (on Sea Ice SIGRID-3, 2014). In the

SIGRID3 format, the sea ice concentration parameter is assigned to the delineated polygons as discrete increments from 0% (0/10) to 100% (10/10), most commonly in steps of 10% (1/10's), and occasionally as intervals, e.g. 40-60% (4/10-6/10). Table 1 shows the SIGRID3 codes for sea ice concentration. In addition to sea ice concentration, the polygons in the ice charts are assigned partial concentrations of sea ice stages of development (e.g. new ice, first-year ice, multiyear ice, etc.) and sea ice floe sizes (e.g. small, big, and giant floes).

ASIDv2+ was generated by spatio-temporally matching regional ice charts from DMI and CIS with Sentinel-1 imagery. Drifting sea ice is dynamic and the sea ice conditions can change drastically within short time timeframes, e.g. within sub-hour time periods. To avoid discrepancies between the sea ice conditions visible in a Sentinel-1 image and the sea ice conditions shown in a regional ice chart, the time difference between the acquisition time of the Sentinel-1 imagery and the timestamp of the ice chart was considered carefully. A restrictive time difference criterion ensures low amounts of drift between the

Sentinel-1 image and the regional ice chart, but it lowers the number of available match-ups that satisfy the criterion, i.e. there is a trade-off between the quality and the quantity of match-ups. In ASIDv2+ the maximum time difference between the





**Table 1.** Descriptions of the SIGRID3 codes used for sea ice concentration in DMI, CIS, and NIC ice charts, as well as the class labels used when training the ConvNets. For a full description of the SIGRID3 format, see (on Sea Ice SIGRID-3, 2014).

| Description | SIGRID3 Code | Class label |
| --- | --- | --- |
| Ice Free | 55 | 0 |
| Less then 1/10 | 01 | 0 |
| Bergy Water | 02 | 0 |
| 1/10 | 10 | 1 |
| 1/10 - 2/10 | 12 | 2 |
| 1/10 - 3/10 | 13 | 2 |
| 2/10 | 20 | 2 |
| 2/10 - 3/10 | 23 | 3 |
| 2/10 - 4/10 | 24 | 3 |
| 3/10 | 30 | 3 |
| 3/10 - 4/10 | 34 | 4 |
| 3/10 - 5/10 | 35 | 4 |
| 4/10 | 40 | 4 |
| 4/10 - 5/10 | 45 | 5 |
| 4/10 - 6/10 | 46 | 5 |
| 5/10 | 50 | 5 |
| 5/10 - 6/10 | 56 | 6 |
| 5/10 - 7/10 | 57 | 6 |
| 6/10 | 60 | 6 |
| 6/10 - 7/10 | 67 | 7 |
| 6/10 - 8/10 | 68 | 7 |
| 7/10 | 70 | 7 |
| 7/10 - 8/10 | 78 | 8 |
| 7/10 - 9/10 | 79 | 8 |
| 8/10 | 80 | 8 |
| 8/10 - 9/10 | 89 | 9 |
| 8/10 - 10/10 | 81 | 9 |
| 9/10 | 90 | 9 |
| 9/10 - 10/10 | 91 | 10 |
| 10/10 | 92 | 10 |





acquisition times of the Sentinel-1 imagery and the timestamps of the regional ice charts was fixed at 5 and 15 minutes for DMI and CIS regional ice charts, respectively. Assuming a maximum sea ice drift of 30 $cm/s$, 5 minutes correspond to 90 $m$ of sea ice drift, which is close to the spatial resolution of Sentinel-1 when operated in the EW mode. Since CIS uses Sentinel-1

imagery less frequently than DMI the time difference criterion was less strict for CIS to ensure roughly equal representation from DMI and CIS in ASIDv2+.

An additional criterion imposed upon potential matches between Sentinel-1 imagery and regional ice charts during the generation of ASIDv2+ was the amount of *valid* information within the extent bounded by each Sentinel-1 scene. At least 25% of the extent of the Sentinel-1 scene had to consist of either sea ice (SIC > 0%) and open water *or* land and open water. This

criterion discarded match-ups that were entirely open water, and, consequently, reduced the skewness of the label distribution in the dataset, which would have otherwise been very heavily dominated by open water. Open water scenes that also contained land were included to ensure coastal representation in ASIDv2+, regardless of the presence of sea ice. As exemplified in Figure 2, the ice charts were rasterized to match the Sentinel-1 geometry and grid spacing.

### 2.1.3 AMSR2 brightness temperatures

The microwave signatures in C-band SAR imagery show patterns related to sea ice formations, but the discrimination between different sea ice conditions is challenged by ambiguities in backscatter intensities, noise phenomena, and wind-induced roughness on the ocean surface, etc. Such ambiguities can degrade the predictive performance of SAR-based sea ice retrieval algorithms (Stokholm et al., 2022; Khaleghian et al., 2021; Boulze et al., 2020; Karvonen, 2022). Approaches based on multi-sensor data fusion schemes that combine SAR imagery and PMW observations have been shown to yield better predictive

performances on sea ice concentration than purely SAR-based approaches (Malmgren-Hansen et al., 2021; Karvonen, 2017).

The AMSR2 brightness temperatures contained in ASIDv2+ include all available frequencies from the AMSR2 sensor (6.9 $GHz$, 7.3 $GHz$, 10.7 $GHz$, 18.7 $GHz$, 23.8 $GHz$, 36.5 $GHz$, 89.0 $GHz$ ) in H/V polarisation. The spatial resolutions of the AMSR2 channels range from a few kilometers to tens of kilometers depending on the frequency. When resampling the AMSR2 brightness temperatures to the SAR geometry for each Sentinel-1 image during the generation of ASIDv2+, all

available AMSR2 L1b swath products within a 7-hour temporal window from the Sentinel-1 acquisition time were considered. A 7-hour temporal window limited the amount of potential sea ice drift between the Sentinel-1 and AMSR2 acquisitions, while ensuring that there were no missing values in the resampled AMSR2 observations, provided that there were no AMSR2 outages. The AMSR2 L1b swath products that intersected the geographical extent bounded by the respective Sentinel-1 image were resampled to a 2 $km$ grid matching the geometry of the Sentinel-1 image using a Gaussian weighted interpolation in

the *pyresample* Python library (Ptresample developers, 2023). The AMSR2 L1b swath products were resampled in ascending order of temporal proximity to the Sentinel-1 acquisition time. Consequently, if the AMSR2 L1b swath that was closest in time to the Sentinel-1 acquisition time did not fully cover the extent of the Sentinel-1 image, the resampled AMSR2 brightness temperatures became a mosaic of multiple AMSR2 swaths (as was the case in the example shown in Figure 2).





## 2.2 NIC Ice Charts

The U.S. National Ice Center (NIC) produces weekly pan-Arctic sea ice charts in accordance with the SIGRID-3 format (on Sea Ice SIGRID-3, 2014). The ice charts are based on observational data acquired up to five days prior to the issue date of the ice chart, and thus, the ice charts represent the sea ice conditions up to five days prior to the ice chart timestamp. The sea ice concentration in NIC ice charts is mainly given as intervals (see Table 1). In section 4, we use the pan-Arctic NIC ice charts as a qualitative reference when investigating the pan-Arctic generalization of our proposed SIC retrieval.

## 2.3 OSI SAF Sea Ice Concentration


The Ocean and Sea Ice Satellite Application Facility (OSI SAF) produces a blended daily L3 sea ice concentration product (OSI-408-a,EUMETSAT (2023)) using traditional Bootstrap and Bristol algorithms (Baordo et al., 2023) on AMSR2 atmospherically corrected brightness temperatures. In section 4, we compare our proposed SIC retrieval to the OSI SAF L3 product at a pan-Arctic scale.

## 3 Methodology


### 3.1 Data preparation

ASIDv2+ is separated at scene level into training, validation, and test splits, containing 5292, 40, and 50 scenes, respectively. The validation and test sets are separated from the training set for the calibration and initial evaluation of our proposed SIC retrieval. The scenes in the validation and test splits are selected specifically to represent both the Greenland and Canadian ice

services, different geographical regions, all seasons, a variety of sea ice conditions, and different wind conditions.

As initially suggested by Malmgren-Hansen et al. (2021), who trained a CNN to predict SIC using the first version of the ASID datasets (ASIDv1) (Malmgren-Hansen et al., 2020), the Sentinel-1 imagery is down-sampled to an $80\,m$ grid spacing, which is better aligned with the spatial resolution of the Sentinel-1 sensor when operated in the EW mode. This effectively increases the spatial extent of the receptive field of the ConvNet without increasing computational costs. It has been sug-

gested that large receptive fields improve the predictive performance of ConvNets in the task of SIC mapping in SAR imagery (Malmgren-Hansen et al., 2021; Stokholm et al., 2022). The AMSR2 brightness temperatures and the ice charts are re-sampled to the same $80\,m$ grid spacing. The scenes are cropped into 1024x1024 patches with 25% overlap between adjacent patches. During training, 512x512 crops are randomly sampled from the 1024x1024 patches. The Sentinel-1 HH/HV bands and the AMSR2 brightness temperatures are standardized prior to training.

The label data, i.e. sea ice concentration maps, are extracted from the SIGRID3 codings in the ice charts using the SIGRID3-to-class-label conversion shown in table 1. Here, each class label represents an assumed sea ice concentration increment, i.e. class label 0 corresponds to a sea ice concentration of 0%, class label 1 corresponds to a sea ice concentration of 10%, etc. The vast majority of the ice chart polygons in ASIDv2+ have been assigned a sea ice concentration in a multiple of 10%, but in the rare case of a polygon being assigned an interval, the mean of the interval is used as the class label, e.g. an interval of 4/10-6/10



in the ice chart is converted to class label 5. If the mean of the interval does not result in a multiple of 10%, we conservatively use the higher bound of the interval as the sea ice concentration increment, e.g. 8/10-9/10 is converted to class label 9.

## 3.2 ConvNet architecture

The ConvNet we employ in this study follows a modified U-Net (Ronneberger et al., 2015) structure with a 6-stage encoder that derives multi-scale features and a decoder that aggregates the information derived from the encoder stages. The ConvNet takes

as input crops of the Sentinel-1 HH/HV bands with concatenated AMSR2 brightness temperatures and outputs non-normalized class scores for each of the sea ice concentration classes (see Table 1) at pixel level, i.e. at 80 $m$ grid spacing. The main building block of the ConvNet is a variant of the inverted residual block (Sandler et al., 2018) with a depthwise separable convolution following the structure shown in Figure 3. The block consists of a pointwise convolution that expands the channel dimension and embeds the low-dimensional feature maps into a higher-dimensional feature space (with an expansion factor $R$), a 3x3

depthwise convolution followed by BatchNorm (BN) (Ioffe, 2017) and the Gaussian Error Linear Unit (GELU) activation function (Hendrycks and Gimpel, 2016), and, finally, a pointwise convolution that projects the high-dimensional feature maps back to a low-dimensional feature space, followed by a LayerScale (LS) (Touvron et al., 2021) operation. The design of the block as well as the macro-structure of the architecture largely follows the findings of Sandler et al. (2018) and Liu et al. (2022).

The spatial down-sampling in the encoder network is carried out using a strided convolution in the residual block at the

beginning of each stage. The spatial up-sampling in the decoder network is carried out using bilinear interpolation prior to each decoder stage. An overview of the ConvNet architecture can be found in Table 2.

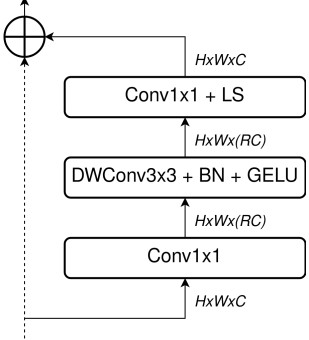

**Figure 3.** Structure of the inverted residual block used in the ConvNet.

## 3.3 Uncertainty quantification

In order for sea ice retrievals provided by deep learning-based retrieval algorithms to be used in safety-critical applications, such as in the context of operational sea ice charting and sea ice information dissemination, or for data assimilation into

numerical ocean-sea-ice forecast models, it is a prerequisite that the retrievals are accompanied by meaningful uncertainties.





**Table 2.** The table describes the architecture of the ConvNet used in this study. Each encoder/decoder stage comprises inverted residual blocks of the variant shown in figure 3, repeated $n$ times. All blocks in the same stage have the same number of output channels $c$. In the encoder, strided $s$ convolutions are used for spatial down-sampling and an expansion factor $R$ is applied to each block. Bilinear interpolation is used for spatial up-sampling prior to each decoder stage.

| Stage | Input dim | n | c | s | R |
|---|---|---|---|---|---|
| | Encoder | | | | |
| Encoder stage 1 | $512^2$ x 16 | 2 | 24 | 1 | 4 |
| Encoder stage 2 | $512^2$ x 24 | 2 | 48 | 2 | 4 |
| Encoder stage 3 | $256^2$ x 48 | 3 | 96 | 2 | 4 |
| Encoder stage 4 | $128^2$ x 96 | 3 | 192 | 2 | 4 |
| Encoder stage 5 | $64^2$ x 192 | 9 | 384 | 2 | 4 |
| Encoder stage 6 | $32^2$ x 384 | 3 | 768 | 2 | 4 |
| | Decoder | | | | |
| Decoder stage 5 | $32^2$ x 1152 | 2 | 384 | 1 | 1 |
| Decoder stage 4 | $64^2$ x 576 | 2 | 192 | 1 | 1 |
| Decoder stage 3 | $128^2$ x 288 | 2 | 96 | 1 | 1 |
| Decoder stage 2 | $256^2$ x 144 | 2 | 48 | 1 | 1 |
| Decoder stage 1 | $512^2$ x 72 | 2 | 24 | 1 | 1 |
| Conv1x1 | $512^2$ x 24 | 1 | 11 | 1 | 1 |

In the following, we consider a classifier with $k$ classes $1, ..., k$. Given an input $\mathbf{x}$, the classifier outputs a $k$-dimensional vector $\mathbf{z}$, often called a *logit*-vector, with non-normalized scores for each class $z_1, ..., z_k$. The logit-vector $\mathbf{z}$ is passed through the softmax function $\sigma$ to obtain class confidence scores $\hat{\mathbf{p}} = \sigma(\mathbf{z})$:

$$\hat{p}_i(y = i|\mathbf{x}) = \frac{exp(z_i)}{\sum_j exp(z_j)}, \qquad i = 1, ..., k \qquad (1)$$

While the confidence vector $\hat{\mathbf{p}}$ upholds the mathematical properties of probabilities, it is empirically known that confidence scores provided by modern neural networks tend to be poorly calibrated (Guo et al., 2017; Lakshminarayanan et al., 2017; Thulasidasan et al., 2020; Ovadia et al., 2019). Here, the calibration of a trained model refers to the accuracy with which the confidence scores provided by the model reflect its predictive uncertainty. If a trained model is poorly calibrated, the confidence scores provided by the model cannot be interpreted as posterior probabilities, nor be used directly to derive meaningful

uncertainties. In this study, we apply and evaluate several strategies to *recalibrate* the class confidence scores provided by our ConvNet, and ultimately use the calibrated confidence scores to derive SIC as well as uncertainties on the SIC estimates.





### 3.3.1 Miscalibration metrics

One notion of miscalibration is the difference in expectation between the accuracy and the confidence of the predictions provided by a trained model, and a popular metric used to quantify this miscalibration is the *Expected Calibration Error* (ECE)
(Naeini et al., 2015). The ECE is computed by partitioning confidence scores into $M$ equal-width bins and taking a weighted average of the difference between the average predicted confidence and the accuracy within each bin:

$$ECE = \sum_{m=1}^{M} \frac{n_m}{N} |acc(B_m) - conf(B_m)| \tag{2}$$

where $acc(B_m)$ is the proportion of correct predictions in bin $B_m$, $conf(B_m)$ is the average predicted confidence in bin $B_m$, $n_m$ is the support of bin $B_m$ and $N$ is the total number of samples.

A vanishing ECE does, however, not always imply a well-calibrated classifier. For instance, the ECE does not necessarily reflect the degree of calibration for individual classes in a multi-class classification scenario (Kull et al., 2019), and the ECE can especially be misleading as a measure of class-wise miscalibration when evaluated on a class-imbalanced dataset. For example, if a very high proportion of all samples comes from a majority class, a minority class will have a negligible contribution to the overall ECE. To mitigate this issue, an averaged *class-wise* ECE (denoted *cwECE* in this study) has been proposed as a
measure of class-wise calibration (Kull et al., 2019):

$$cwECE = \frac{1}{k} \sum_{i=1}^{k} \sum_{m=1}^{M} \frac{n_{m,i}}{N} |acc(B_{m,i}) - conf(B_{m,i})| \tag{3}$$

Here, the ECE is computed for each class individually, and finally, the class-wise ECE's are averaged to obtain the cwECE.

Similarly, as the contribution of each confidence bin to the overall ECE is weighted by the bin support, the ECE measure will be dominated by over-represented confidence regions. If a very high proportion of all samples have a high predicted
confidence, the lower - or mid-confidence regions will have a negligible contribution to the overall ECE, and the ECE will not be informative on the under-represented confidence regions. To weigh the calibration error equally across confidence regions, a *region-balanced* ECE (denoted *rbECE* in this study) has been proposed (Dawkins and Nejadgholi, 2022):

$$rbECE = \frac{1}{n_\Theta} \sum_{B_m \in \Theta} |acc(B_m) - conf(B_m)| \tag{4}$$

Here, the calibration error for each bin contributes equally to the overall calibration error. However, to ensure that $acc(B_m)$ is
well-defined, it is subject to a bin support threshold $n_m > t_\Theta$. The set of bins that meet the bin support requirement is denoted by $\Theta$ and the number of bins in $\Theta$ is denoted by $n_\Theta$.

We combine the cwECE and the rbECE to obtain an averaged class-wise region-balanced ECE, which we denote *cwrbECE*:





$$cwrbECE = \frac{1}{k} \sum_{i=1}^{k} \frac{1}{n_\Theta} \sum_{B_{m,i} \in \Theta} |acc(B_{m,i}) - conf(B_{m,i})| \tag{5}$$

Here, we compute the rbECE separately within each class and average the class-wise contributions. While the cwrbECE
accounts for some insufficiencies in the ECE, it is still sensitive to biases introduced by binning, the finite sample size when
computing the per-bin statistics, and - in extension - the choice of the bin support threshold.

Complementary to a quantitative assessment of miscalibration, the calibration of a classifier can be assessed qualitatively
by examining reliability diagrams (Murphy and Winkler, 1977). Reliability diagrams plot the per-bin predictive accuracy as a
function of the per-bin predictive confidence. A reliability diagram for a perfectly calibrated classifier will show the identity
function, while any deviation from the diagonal represents miscalibration.

### 3.3.2 Recalibration strategies

Many strategies have been proposed to improve the calibration of deep learning models. In our study, we evaluate three
approaches to improving the calibration of our proposed SIC retrieval; *parametric rescaling of logits* (Guo et al., 2017), *label
smoothing* (Szegedy et al., 2015) and *deep ensembling* (Lakshminarayanan et al., 2017).

Post-hoc parametric scaling was initially proposed within the context of neural network calibration by Guo et al. (2017).
These scaling approaches use a hold-out validation set to learn a single parameter or a set of parameters, to rescale the logit
vector $\mathbf{z}$ before passing $\mathbf{z}$ through the softmax function. In the case of temperature scaling, a single learned scalar, the *temper-
ature* ($T$), is used to either raise ($T < 1$) or lower ($T > 1$) the confidence $\hat{\mathbf{p}} = \sigma(\mathbf{z}/T)$. However, in a multi-class classification
setting, the degree and direction of miscalibration might vary between classes (e.g. the predicted confidence generally being
over-confident for one class, and under-confident for another class), and it might be beneficial to learn a separate temperature
for each class $\mathbf{T} = T_1, ..., T_k$ (Frenkel and Goldberger, 2021). In this case, which we denote *class-wise* temperature scaling, the
logits are rescaled using a set of learned class-wise temperatures $\hat{\mathbf{p}} = \sigma(\mathbf{z}/\mathbf{T})$. In vector scaling, multiple parameters ($\mathbf{W}, \mathbf{b}$) are
learned to perform a linear transformation of the logit vector $\mathbf{z}$ before passing it through the softmax function $\hat{\mathbf{p}} = \sigma(\mathbf{W}\mathbf{z} + \mathbf{b})$.
Here, the off-diagonal elements of $\mathbf{W}$ are fixed to zero (Guo et al., 2017). For temperature scaling, class-wise temperature and
vector scaling, the associated parameters T, $\mathbf{T}$ and $\mathbf{W}$, $\mathbf{b}$ are learned by optimizing the negative log-likelihood on the held-out
validation set.

*Label smoothing* (LS) was originally proposed as a regularization technique to improve the generalization of ConvNets for
image classification by Szegedy et al. (2015). With label smoothing, the target becomes a mixture of the one-hot encoded label
and a uniform distribution with a smoothing factor $\epsilon$, transforming the one-hot encoded label from 0 and 1 to $\frac{\epsilon}{k-1}$ and $1 - \epsilon$,
respectively. It has been suggested that by artificially softening the targets, label smoothing implicitly calibrates the confidence
scores of neural networks (Müller et al., 2020).

Lastly, it has been observed that ensembling of neural networks (i.e. the averaging of the predictive confidence scores $\hat{\mathbf{p}}$
from multiple trained neural networks) not only improves predictive performance (Lakshminarayanan et al., 2017; Allen-
Zhu and Li, 2020), but also predictive uncertainty estimation (Lakshminarayanan et al., 2017; Wen et al., 2020). It has been



demonstrated that the random initialization of the neural network parameters as well as data shuffling during training introduce the ensemble diversity necessary to obtain a significantly improved uncertainty quality, even with ensembles consisting of as few as 5 ensemble members (Lakshminarayanan et al., 2017; Ovadia et al., 2019).

### 3.3.3 Retrieving SIC from calibrated class confidence scores

In a multi-class classification setting it is common practice to retrieve the predicted class using $\hat{y}_i = argmax(\hat{\mathbf{p}})$ and the associated class confidence using $\hat{p}_i = max(\hat{\mathbf{p}})$. This practice has been adopted in recent SIC classification studies (Stokholm et al., 2022; Iris et al., 2021). While this approach does indeed retrieve the SIC class deemed the most likely by the classifier, as well as the associated confidence, it completely disregards the remaining information provided in the confidence vector.

Alternatively, given a classifier that outputs *well-calibrated* confidence scores, we propose to retrieve the pixel-wise $SIC$ and its associated uncertainty $\sigma_{SIC}$ as a weighted average and a weighted standard deviation of the 11 SIC increments $\mathbf{I} = 0, 10, ..., 100$ (denoted by the class labels in table 1), respectively, with weights given by their respective confidence scores $\hat{\mathbf{p}}$ provided by the classifier:

$$SIC = \sum_i \hat{p}_i I_i \qquad i = 1, ..., k \tag{6}$$

$$\sigma_{SIC} = \sqrt{\sum_i \hat{p}_i (I_i - SIC)^2} \qquad i = 1, ..., k \tag{7}$$

Contrary to the conventional $argmax(\hat{\mathbf{p}})/max(\hat{\mathbf{p}})$-approach, our proposed SIC retrieval exploits the entirety of the information provided by the classifier. Equations 6 and 7 result in a continuous SIC output and an associated uncertainty characterized by a standard deviation, as opposed to a discrete SIC output, with SIC increments mirroring the manually produced ice charts, and an associated confidence.

## 3.4 Experimental setup

We train a total of 10 ConvNets, 5 with label smoothing ($\epsilon = 0.1$) and 5 without label smoothing ($\epsilon = 0$). We train multiple ConvNets to report averaged miscalibration metrics with accompanying standard deviations accounting for the effect of the stochasticity introduced during the training of the ConvNets as well as the random initializations of the network parameters. We train all ConvNets with mixed precision for 45 epochs using the AdamW (Loshchilov and Hutter, 2017) optimizer with an initial learning rate of $3e-4$ and a multi-step learning rate scheduling policy to lower the learning rate sequentially as training progresses. We use cross-entropy as the loss function. We use a batch size of 24 and train the ConvNets from scratch on a single NVIDIA A100. For data augmentation, we adopt common augmentation techniques including random cropping, random horizontal and vertical flipping, random scaling, and random rotation. We regularize the ConvNets with Stochastic Depth (Huang et al., 2016).





For each trained ConvNet, we apply three different parametric rescaling techniques for recalibration: temperature scaling, class-wise temperature scaling, and vector scaling. The associated scaling parameters T, $\mathbf{T}$ and $\mathbf{W}$, $\mathbf{b}$ are learned through

optimization of the negative log-likelihood on the held-out ASIDv2+ validation set. We conduct this optimization in both unweighted and weighted settings. In the weighted setting, we weigh the class-wise contributions to the negative log-likelihood loss by the inverse of the class proportions in the ASIDv2+ training set. Furthermore, we explore the effect of ensembling on the calibration quality for each parametric scaling technique. The ensembling is carried out by averaging the predicted confidence scores from 5 ConvNets. When computing the ECE and cwrbECE metrics based on the ASIDv2+ test set, we partition the pre-

dicted confidence scores into 10 equally-spaced bins from 0-100%. For the cwrbECE metric, we set to bin support requirement to $1e6$. In addition to the quantitative assessment of the calibration quality, we carry out a qualitative assessment of the recalibration strategies with the lowest miscalibration according to the ECE and cwrbECE scores by examining reliability diagrams based on the ASIDv2+ test set. Based on the quantitative and qualitative assessment of the calibration quality, we identify the most effective recalibration strategy, and apply said strategy in the subsequent analysis of the predictive performance of our

SIC retrieval.

We perform an initial quantitative predictive performance evaluation of our SIC retrieval against regional ice charts from the ASIDv2+ test set. To account for class imbalance, we use weighted RMSE as the summarizing measure of the predictive performance, weighing samples from each class by the inverse of the class proportions in the ASIDv2+ test set. We include a visual qualitative assessment of the predictive performance for a subset of the scenes in the ASIDv2+ test set.

To investigate the predictive performance of our SIC retrieval at a pan-Arctic scale, we compare pan-Arctic mosaics of our SIC retrievals generated from 7-days worth of Sentinel-1 EW imagery to mosaics of the OSI SAF SIC product introduced in section 2.3. We choose 7-day periods to ensure decent spatial representation of most of the Arctic in the Sentinel-1 coverage. For each Sentinel-1 scene in the 7-day period, we resample the OSI SAF product that best aligns temporally with the Sentinel-1 acquisition time to the geographical extent delineated by said Sentinel-1 scene on a polar stereographic grid. We put the newest

data on top in the resulting mosaic. We use the $R^2$-score and the mean bias between our SIC retrieval and OSI SAF to quantify the discrepancies between the products. These summarizing statistics are evaluated at the OSI SAF grid spacing of 12.5 $km$. However, the pan-Arctic plots shown in the following sections are generated at 1 $km$ grid spacing to allow the reader to study the accompanying SAR scenes as well as the differences in spatial resolution between our SIC retrieval and OSI SAF. As an additional reference, we show pan-Arctic ice charts produced by NIC with an issue date within the 7-day period. Note, however,

that there can be a lag of several days (up to 12 days in the worst case) between the acquisition time of the observations used to generate the mosaics and the acquisition time of the observations used to produce the NIC ice chart.

The loss of Sentinel-1B on December 23rd, 2021 severely restricted the acquisition of Sentinel-1 images in the Arctic region, with no images being acquired in the central Arctic at all. To ensure central Arctic coverage in the comparative study of our SIC retrieval against OSI SAF, we choose three 7-day periods in 2020, when Sentinel-1A and Sentinel-1B both were

in operation. Sentinel-1 scenes contained within the mosaics for the selected 7-day periods in 2020 that are also part of the ASIDv2+ training or validation sets are not included in the computation of the summarizing statistics. To demonstrate the generalization of our SIC retrieval to contemporary observations, we include three 7-day mosaics from 2023 as well. Although



the Arctic coverage is sparse, with only Sentinel-1A being in operation, it is essential to investigate the generalization of our SIC retrieval to observations outside of the temporal bounds of the ASIDv2+ dataset on which the ConvNets are trained.

For each year, we select a 7-day period in the freezing season (October-March), in the melting season (April-September), and around the yearly minimum (August/September/October). Apart from this criterion, the 7-day periods are chosen arbitrarily, only subject to the availability of pan-Arctic NIC ice charts.

## 4    Results

### 4.1    Calibration

Table 3 contains the ECE and cwrbECE metrics evaluated against the ASIDv2+ test set for the recalibration strategies presented in section 3.3.2. The lowest miscalibration scores are obtained by ensembles of ConvNets trained without label smoothing ($\epsilon = 0$). The lowest ECE score is obtained by an ensemble of temperature scaled ConvNets, whereas the lowest cwrbECE score is obtained by an ensemble of vector scaled ConvNets. Note the relatively large variations in the ECE and cwrbECE scores between the 5 ConvNets, indicating that the random network parameter initialization as well the stochasticity introduced the

during the training of the ConvNets (e.g. from data shuffling and data augmentation) can have a non-negligible impact on the final calibration of the ConvNets.

Figure 4 shows reliability curves for the recalibration strategies with the lowest ECE and cwrbECE scores, as well as an ensemble of uncalibrated ConvNets for reference. As evident in Figure 4A, the ensemble of vector scaled ConvNets achieves a low miscalibration error across all confidence regions, while the ensembles of temperature scaled or uncalibrated ConvNets

are under-confident in the mid - and higher confidence regions. Additionally, the class-wise averaged reliability curves in Figure 4B show that the ensemble of vector scaled ConvNets achieves the lowest class-wise miscalibration error across most confidence regions as well. Based on the reliability diagrams, we identify vector scaling as the most effective recalibration strategy in our study and the ensemble of vector scaled ConvNets as the best calibrated model configuration, suggesting that our proposed cwrbECE metric is superior to the standard ECE metric when identifying miscalibration - particularly when

regarding class-wise miscalibration and miscalibration across confidence regions.

From here on, when referring to the *ASIP* SIC retrieval, it is implied that we are referring to our proposed sea ice concentration retrieval from confidence scores $\hat{\mathbf{p}}$ (see equation 6 and 7 in section 3.3.3) provided by an ensemble of vector scaled ConvNets.

### 4.2    SIC retrievals

Figure 5 shows the predictive performance of the ASIP retrieval against the regional ice charts in the ASIDv2+ test set. For all polygons of each sea ice concentration increment in ice charts (see class labels in table 1), we compute the mean and the standard deviation of the sea ice concentrations retrieved at $80\ m$ pixel spacing using the ASIP retrieval. The ASIP retrieval achieves an overall $R^2$-score of $95\%$, with the largest deviations occurring at the intermediate sea ice concentrations. This



**Table 3.** The table summarises the ECE and cwrbECE metrics evaluated against the ASIDv2+ test set for the recalibration strategies presented in section 3.3.2. The lowest ECE and cwrbECE scores are shown in bold.

| Recalibration strategy | ECE ↓ | | cwrbECE ↓ | |
|---|---|---|---|---|
| | Without ensembling | With ensembling | Without ensembling | With ensembling |
| Without LS ($\epsilon = 0$) | | | | |
| Uncalibrated | 0.0034 (0.00025) | 0.0033 | 0.0587 (0.0053) | 0.0637 |
| Temperature scaling | 0.0034 (0.00022) | 0.0033 | 0.0576 (0.0046) | 0.0638 |
| Temperature scaling (weighted) | 0.0033 (0.00026) | **0.0032** | 0.0585 (0.0035) | 0.0574 |
| Class-wise temperature scaling | 0.0035 (0.00027) | 0.0033 | 0.0582 (0.0038) | 0.0577 |
| Class-wise temperature scaling (weighted) | 0.0034 (0.00016) | 0.0033 | 0.0649 (0.0038) | 0.0619 |
| Vector scaling | 0.0037 (0.00022) | 0.0035 | 0.0581 (0.0037) | **0.0539** |
| Vector scaling (weighted) | 0.0038 (0.00015) | 0.0036 | 0.0706 (0.0067) | 0.0668 |
| With LS ($\epsilon = 0.1$) | | | | |
| Uncalibrated | 0.0228 (0.00040) | 0.0232 | 0.0688 (0.0054) | 0.0675 |
| Temperature scaling | 0.0073 (0.00027) | 0.0070 | 0.0675 (0.0036) | 0.0681 |
| Temperature scaling (weighted) | 0.0068 (0.00028) | 0.0067 | 0.0633 (0.0084) | 0.0635 |
| Class-wise temperature scaling | 0.0048 (0.00006) | 0.0047 | 0.0615 (0.0042) | 0.0617 |
| Class-wise temperature scaling (weighted) | 0.0055 (0.00013) | 0.0054 | 0.0601 (0.0054) | 0.0610 |
| Vector scaling | 0.0050 (0.00056) | 0.0048 | 0.0614 (0.0045) | 0.0588 |
| Vector scaling (weighted) | 0.0060 (0.00028) | 0.0059 | 0.0599 (0.0036) | 0.0583 |

behaviour is expected as previous studies on the inter-analyst variation in manually produced ice charts document the largest
disagreement among ice analysts at the intermediate sea ice concentrations (Karvonen et al., 2015; Cheng et al., 2020). The ASIP retrieval achieves similar predictive performances in the freezing and melting seasons, respectively. There is a slight tendency of the ASIP retrieval to overestimate lower sea ice concentrations and underestimate the higher sea ice concentrations when compared to the regional ice charts. The ASIP retrieval achieves an overall weighted RMSE of 15.8%, compared to 20.2% when substituting equation 6 with the conventional *argmax*-approach (see section 3.3.3).

Figure 6 shows five examples from the ASIDv2+ test set of Sentinel-1 HH imagery, sea ice concentration from the manually produced regional ice charts, and sea ice concentrations retrieved using the ASIP retrieval with their associated uncertainties. All examples are depicted in the original Sentinel-1 SAR geometry and the geographical extent of each example is outlined in Figure 1. Generally, the ASIP retrieval produces sea ice maps that resemble the manually produced ice charts to a significant extent. However, as the manually produced ice charts are drawn as smooth delineated polygons of relatively homogeneous
sea ice conditions, the sea ice maps produced using the ASIP retrieval might contain more detail and variability, with a larger degree of similarity to the spatial patterns and textural intricacies in underlying SAR imagery.



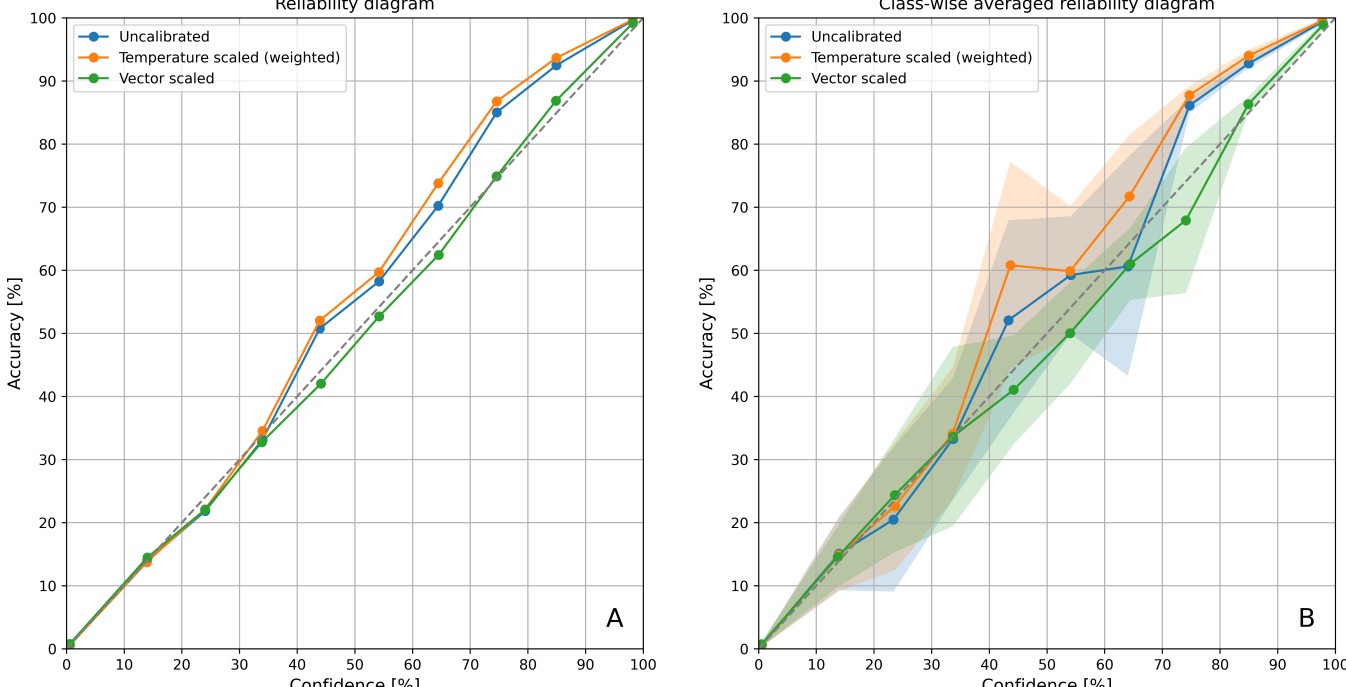

**Figure 4. A**: Reliability curves for ensembles of uncalibrated (*blue*), temperature scaled (*orange*) and vector scaled (*green*) ConvNets. **B**: Class-wise averaged reliability curves for ensembles of uncalibrated (*blue*), temperature scaled (*orange*) and vector scaled (*green*) ConvNets. Evaluated on the ASIDv2+ test set.

Figure 6A-D and Figure 6E-H show examples from the melting season with a diverse range of sea ice concentrations. Figure 6A-D is a May scene from the mouth of the Hudson Strait showing the remains of the southerly transport of predominantly first-year ice formed in Baffin Bay during the freezing season. In Figure 6E-H from the Canadian Arctic Archipelago at the

height of the melting season, we see the microwave signatures in the SAR imagery of near-coastal low and intermediate sea ice concentrations being well-interpreted by ASIP. In both examples, we generally see the highest uncertainties at the intermediate sea ice concentrations, and very low uncertainties in regions of open water or densely packed sea ice. In Figure 6I-L from the Scoresbysund Fjord in East Greenland, densely packed drift ice consisting predominantly of multiyear ice from the Arctic Ocean is being transported south along the eastern coast of Greenland. The distinct edge of the ice pack is reproduced in the

ASIP retrieval, and the low-backscatter region of smooth land fast ice within the fjord system, which is known to be difficult to accurately map in SAR-based sea ice retrievals (Stokholm et al., 2022; Khaleghian et al., 2021), is correctly recognized as high concentration sea ice with a high certainty. In 6M-P the wind-roughening of the ocean surface leads to very high backscatter intensities over open water, particularly in the near - to mid-range, which consequently leads to the sea ice in the mid - to far-range exhibiting relatively low backscatter intensities in the resulting SAR image. Strong wind conditions can complicate

the interpretation of the microwave signatures in the SAR-based sea ice retrievals (Malmgren-Hansen et al., 2021; Wang and



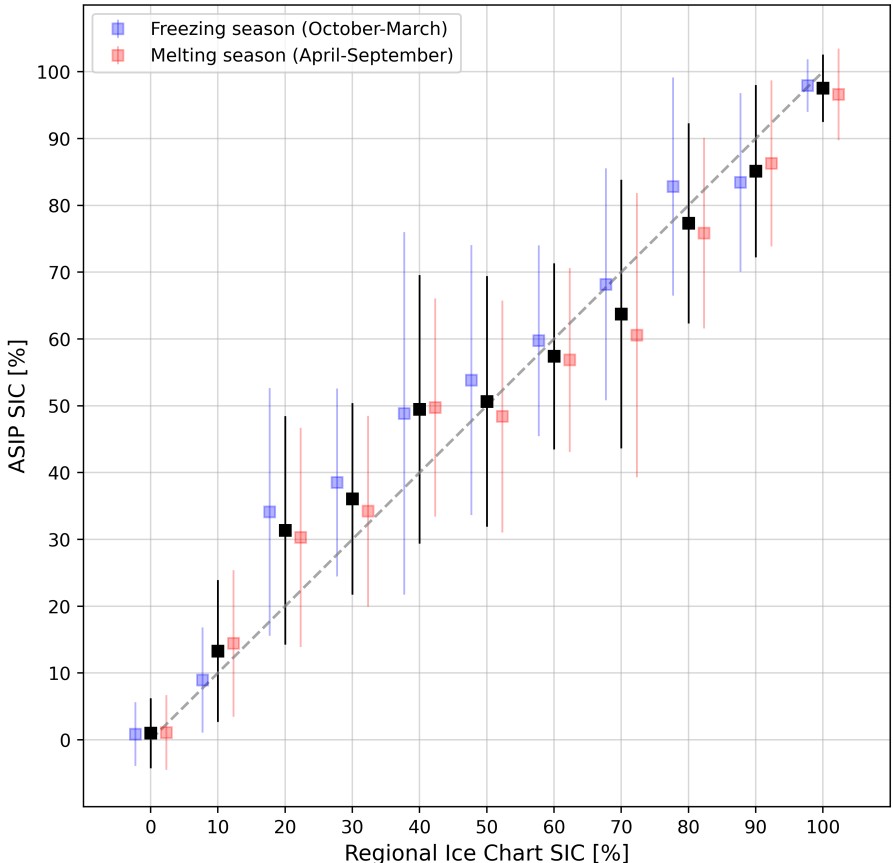

**Figure 5.** Predictive performance of the ASIP retrieval against the regional ice charts in the ASIDv2+ test set. For each sea ice concentration increment (e.g. 0%, 10%, 20%, ..., 100%, see table 1), the mean and the standard deviation of the sea ice concentrations retrieved by ASIP are computed. The red and blue squares correspond to the aggregated results from samples acquired during the melting and freezing seasons, respectively, while the black squares correspond to the aggregated results from all samples.

Li, 2021). While ASIP handles this case fairly well, the sea ice concentrations in the ice zone adjacent to the ice edge are being overestimated compared to the manually produced ice chart. Lastly, Figure 6Q-T is an October scene from the northern Baffin Bay with formations of new and young ice. Newly formed sea ice can take a number of ice forms depending on the sea state, temperature, and wind conditions at the time of ice formation, which can lead to ambiguous microwave signatures in the resulting SAR imagery. The ambiguous microwave signatures of newly formed sea ice can ultimately lead to high uncertainties in the ASIP retrieval, as exemplified in Figure 6T.

Figures 7 and 8 show examples from 2020 and 2023, respectively, of the ASIP retrieval applied at a pan-Arctic scale to produce 7-day mosaics of sea ice concentration. The mosaics clearly show the tendency of the ASIP retrieval to be very certain in regions of open water and densely packed sea ice, while being the most uncertain in the marginal ice zone.



**Figure 6.** 5 examples scenes from the ASIDv2+ test set. From left to right: Sentinel-1 HH, manually produced regional ice chart from CIS or DMI, sea ice concentration retrieved by ASIP, uncertainty reported by ASIP. Zoom in to view details.





**Figure 7.** Three 7-day mosaics from 2020 of sea ice concentration retrieved using the ASIP retrieval. From left to right: Sentinel-1 HH mosaic, ASIP sea ice concentration mosaic, and ASIP uncertainty mosaic. Zoom in to view details.

## Comparative Analysis of ASIP and OSI SAF in the Pan-Arctic region

In Figures 9 and 10, we conduct a comparison between the ASIP retrieval and OSI SAF for the same mosaics that were presented in Figures 7 and 8. A qualitative look at the NIC ice charts, ASIP, and OSI SAF reveals the sea ice extents in the







**Figure 8.** Three 7-day mosaics from 2023 of sea ice concentration retrieved using the ASIP retrieval. From left to right: Sentinel-1 HH mosaic, ASIP sea ice concentration mosaic, and ASIP uncertainty mosaic. Zoom in to view details.

respective products to be quite similar. The last columns of Figures 9 and 10 show difference maps between ASIP and OSI SAF, enabling investigations into the spatial distributions of concentration biases, with the $R^2$-score and mean bias provided as summarizing statistics. The $R^2$-scores are generally high (>90%), indicating a decent level of agreement between ASIP and






OSI SAF. This high level of agreement, coupled with the evident similarity in Arctic sea ice extent across the three products, indicate a robust generalization capability in the ASIP retrieval, and importantly, that this generalization extends well beyond the geographical and temporal bounds of the training dataset. There is, however, a clear general tendency of the ASIP retrievals to contain more sea ice than OSI SAF, with mean biases ranging from 1.45% to 8.55% for the six example mosaics. These

biases predominantly stem from discrepancies in the marginal ice zone, where the ASIP retrievals contain considerably more sea ice than OSI SAF.

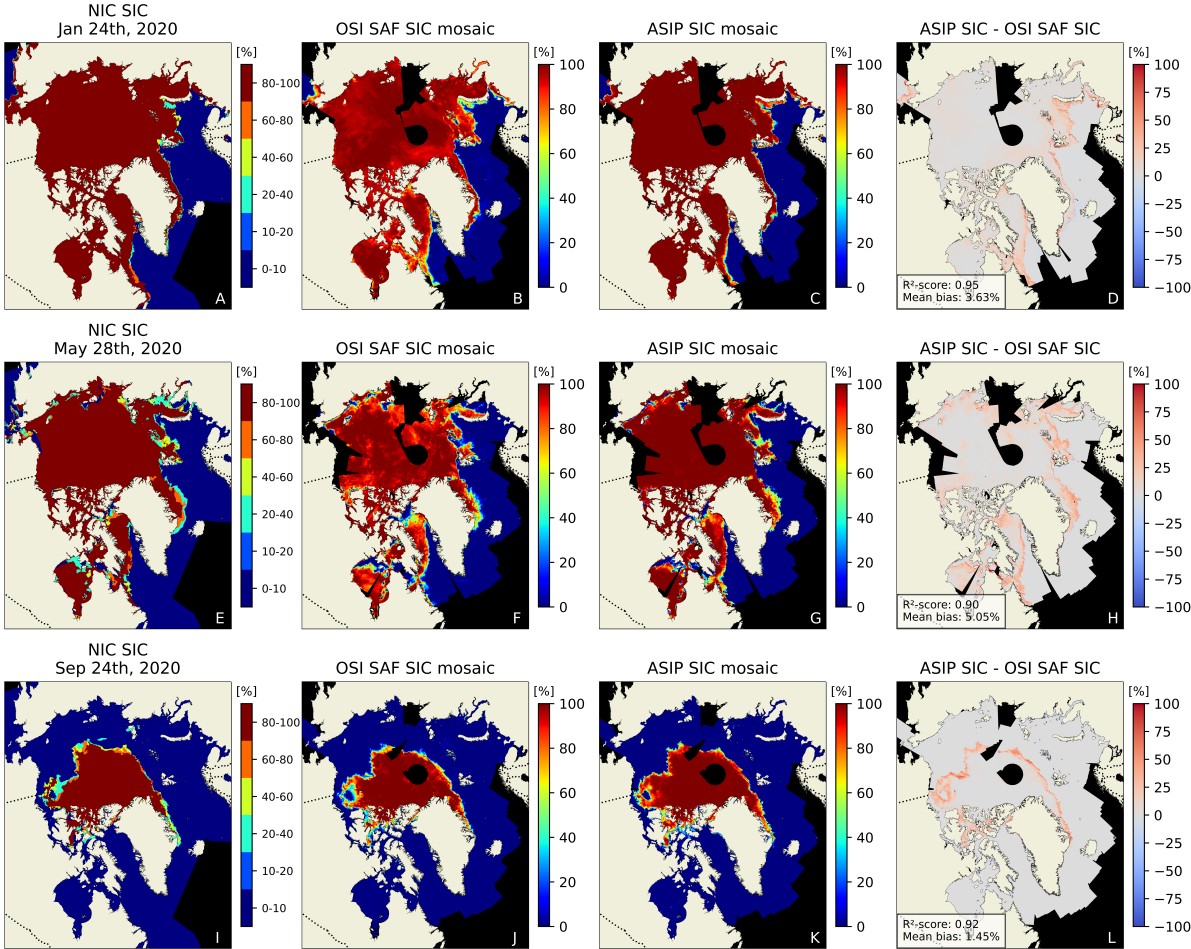

**Figure 9.** Pan-Arctic comparisons between ASIP and OSI SAF for three 7-day periods in 2020. From top to bottom, 7-day periods are; Jan 20th to Jan 27th, May 25th to June 1st, and Sep 21st to Sep 28th. NIC ice charts are included as an additional reference. From left to right; NIC ice chart, OSI SAF SIC mosaic, ASIP SIC mosaic, difference map between ASIP SIC and OSI SAF SIC. The corresponding Sentinel-1 HH mosaic can be seen in Figure 7. Zoom in to view details.

In Figure 10I-L from the first week of August 2023, the mismatch in sea ice concentration between ASIP and OSI SAF is particularly pronounced in the Canadian Archipelago and the Russian High Arctic. A zoomed-in view spanning the region





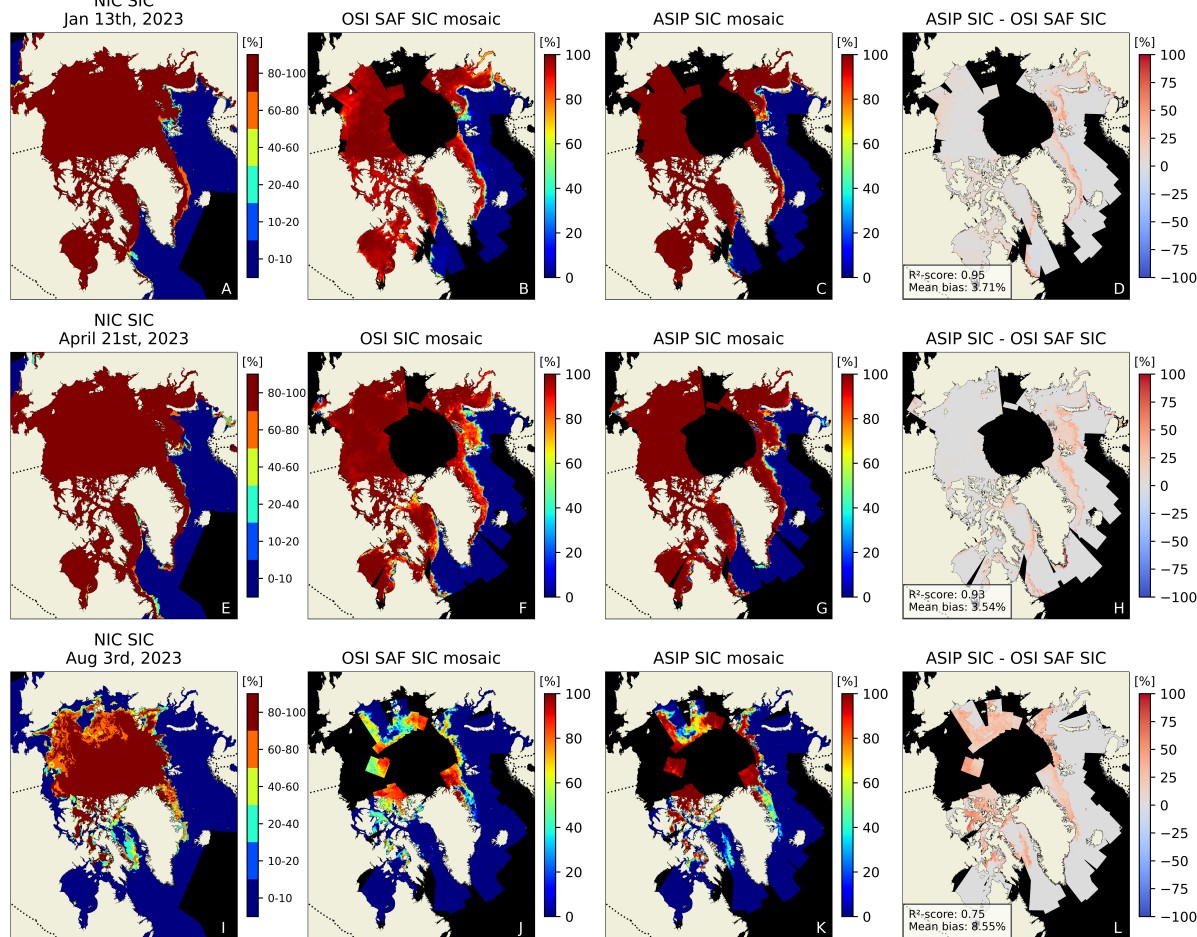

**Figure 10.** Pan-Arctic comparisons between ASIP and OSI SAF for three 7-day periods in 2020. From top to bottom, 7-day periods are; Jan 9th to Jan 15th, April 17th to April 23rd, and July 31st to Aug 6th. NIC ice charts are included as an additional reference. From left to right; NIC ice chart, OSI SAF SIC mosaic, ASIP SIC mosaic, difference map between ASIP SIC and OSI SAF SIC. The corresponding Sentinel-1 HH mosaic can be seen in Figure 7. Zoom in to view details.

from Franz Josef Land toward the Kara Sea with the accompanying Sentinel-1 imagery is shown in Figure 11. While the extent

of the sea ice is fairly similar in ASIP and OSI SAF, the sea ice concentrations are significantly higher in the ASIP retrieval. Furthermore, ASIP is able to map the sea ice with a much higher level of detail, capturing small-scale radiometric and spatial variation in the underlying SAR imagery.

Figure 12 shows an example from the Baffin Bay from August 2023. Here, a visual comparison between OSI SAF and the Sentinel-1 imagery reveals that OSI SAF is missing large chunks of the broken-up sea ice in Baffin Bay as well as sea ice in

the near-coastal regions, which is especially prevalent along the western coast of Baffin Island. Contrarily, ASIP recognizes




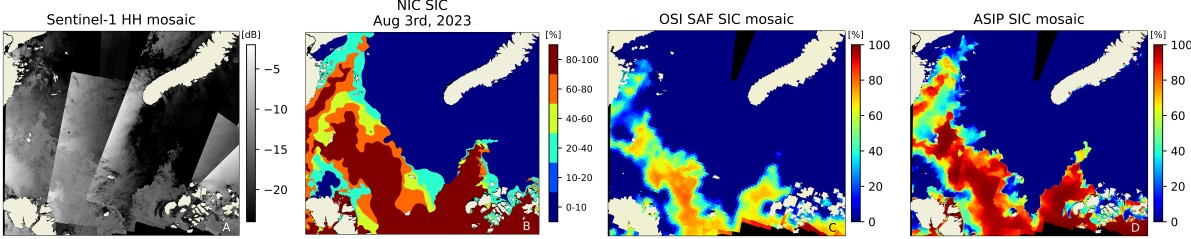

**Figure 11.** Zoomed-in view of Figure 10I-L spanning the region from Franz Josef Land toward the Kara Sea. From left to right; Sentinel-1 HH mosaic, NIC ice chart, OSI SAF SIC mosaic and ASIP SIC mosaic.

the rather ambiguous microwave signatures from the central Baffin Bay in the SAR imagery as low-to-intermediate sea ice concentrations, while providing a detailed mapping of the sea ice in the coastal regions as well.

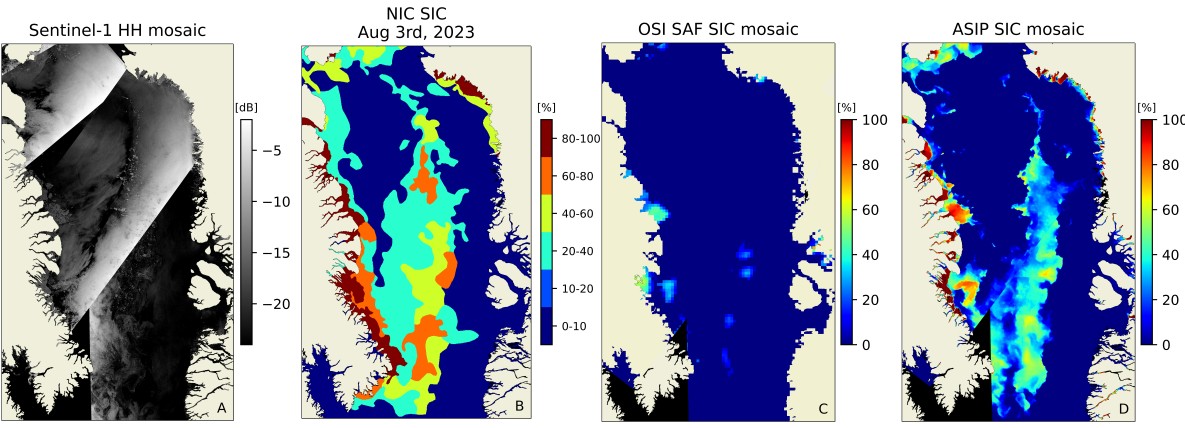

**Figure 12.** Zoomed-in view of Figure 10I-L covering Baffin Bay. From left to right; Sentinel-1 HH mosaic, NIC ice chart, OSI SAF SIC mosaic and ASIP SIC mosaic.

## 5    Discussions

While ASIP offers the potential to provide more detailed representations of the sea ice cover than many existing satellite-based
sea ice products, it also has limitations that are inherent to the ASIP retrieval methodology. These deficiencies can arise from ambiguities and unwanted signals in the input observations, or from the training and calibration procedure, e.g. the choice of manually produced ice charts as "ground truth" data.

Despite the notable advantage of ASIP's high spatial resolution, enabling SIC retrievals in coastal areas, ASIP at times struggles to confidently map narrow fjords. The contextual information derived from Sentinel-1 SAR observations in narrow
fjords tends to be fraught with ambiguity, with barely any textural information. Moreover, the issue is exacerbated by land





spill-over effects caused by the large footprints in the brightness temperature observations from AMSR2, potentially leading to erroneous sea ice concentrations in the ASIP retrievals. Additionally, ASIP can be susceptible to errors stemming from ambiguous backscatter signatures in the Sentinel-1 SAR imagery in coastal as well as offshore regions. Such ambiguities can occur for a variety of reasons. For instance, when encountering newly formed sea ice that forms as a thin ice film on the ocean

surface, which may resemble the open ocean in calm wind conditions, or instances of high backscatter intensities brought on by wind-induced capillary waves, ASIP may generate inaccurate sea ice retrievals and report high uncertainties. Furthermore, ambiguities or unwanted signals in the AMSR2 brightness temperatures, such as certain atmospheric contributions to the signal or the presence of melt ponds on top of the sea ice, can lead to a similar degradation of the ASIP output.

There are both advantages and disadvantages associated with the use of manually produced ice charts as label data for

the training and calibration of the ASIP retrieval. Most importantly for this application, ice charts are produced operationally at multiple national ice services, ensuring the widespread availability of ice charts, spanning vast geographical areas, and capturing the seasonal variations of the sea ice. Such comprehensive and diverse label data, with rarely occurring sea ice conditions represented, is crucial for developing a robust model that is suitable for operational use and capable of generalizing beyond the geographical and temporal bounds of the label data on which it was trained. Further, manually produced ice charts

have high spatial resolutions compared to other available sea ice products, such as products derived from passive microwave observations, particularly along the sea ice edge. The ice charts are often produced on the basis of SAR images, enabling the compilation of training datasets consisting of very timely - if not exact - match-ups between ice chart and SAR image, which is important due to the high spatial resolution of the SAR sensor and the continuous movement of drifting sea ice. However, as ice charts are drawn by ice analysts by manual interpretation of satellite observations, there are bound to be inherent uncertainties

in the ice charts, such as analyst subjectivity, and inter - and intra-analyst variability (Karvonen et al., 2015; Kreiner et al., 2023). While such uncertainties are recognized in the ice charting community (International Ice Charting Working Group, 2021), they are neither quantified nor conveyed to users. Any systematic biases introduced during the ice charting process may propagate into the ASIP retrieval results. For instance, national ice services prioritize the safety of maritime vessels, potentially leading ice analysts to adopt a *rather-too-much-ice-than-too-little-ice*-mentality when delineating ice polygons and assigning

sea ice concentrations. Consequently, there may be an overestimation of intermediate sea ice concentrations in the ice charts. If such a tendency exists within the ice services, it is likely to be reproduced in the ASIP retrievals. A comparative analysis was conducted at a pan-Arctic scale to assess the sea ice retrievals from ASIP in comparison to OSI SAF. The results reveal that the ASIP retrieval consistently exhibits relatively higher sea ice concentration estimates, with mean biases ranging from 1.45% to 8.55% for the chosen time periods. The difference maps presented in Figures 9 and 10 show that these biases are primarily

attributed to disparities in the marginal ice zone, i.e. at the intermediate sea ice concentrations. While part of these discrepancies may indeed stem from systematic biases in the ice charts learned by ASIP during the training process, the examples shown in Figures 11 and 12 illustrate that OSI SAF on occasion is underestimating intermediate sea ice concentrations.





**Future work**

While this study focuses on sea ice concentration, the ice charts in ASIDv2+ and other datasets in the ASID family (Buus-
Hinkler et al., 2022; Saldo et al., 2020; Malmgren-Hansen et al., 2020) contain partial concentrations of sea ice stage of
development (SoD) and sea ice form (e.g. floe size) in addition to sea ice concentration. An exciting advancement is to utilize
all three sea ice parameters in the ice charts to train deep learning models capable of simultaneous multi-parameter retrieval
(Wulf et al., 2022). Here, one interesting aspect is the potential to improve generalization as well as the predictive performance
for each parameter when training *multi-headed* models performing multiple tasks simultaneously (Caruana, 1997). Further,
the sea ice community has expressed the need for improved SAR-based retrievals of multiple sea ice parameters to be in-
tegrated into ice service routines and for data assimilation purposes (Korosov et al., 2023). Recently, we co-organized the
ESA-sponsored *AutoICE* challenge in which the participants were incentivized to develop deep learning models capable of
multi-parameter retrieval from SAR imagery using the dataset we prepared for the challenge (Buus-Hinkler et al., 2022). For
the maritime community and the national ice services, the SoD is particularly important due to the advisory work of the ice
services depending on information about the distribution SoDs in the user's region of operation and the polar ice class (Interna-
tional Maritime Organization, Maritime Safety Committee, 2016) of the user's vessel. The SoD categories (e.g. new ice, young
ice, thin first-year ice, thick first-year ice and multi-year ice) can be difficult to discriminate accurately in ML-based retrieval
methodologies due to some categories having similar backscatter signatures as well as the ambiguity of having partial con-
centrations as labels (i.e. each pixel containing multiple SoD categories) (Wulf et al., 2023a). When ice analysts assign partial
concentrations of SoDs to the polygons in an ice chart, their analysis is primarily based on the available satellite observations,
but they also rely on their extensive experience and knowledge about the 'typical' (or climatological) sea ice conditions in
specific locations at specific times of the year. If a SAR image is particularly difficult to interpret due to ambiguities in the sea
ice backscatter signatures, the analyst can draw from their experience to resolve those ambiguities. The ConvNets presented in
this study lack this experience and rely solely on satellite observations. Allowing the ConvNets to learn the location-dependent
seasonal variation in sea ice conditions, either by including the location and the time of the year as additional input features or
by some other mechanism, we can level the playing field between the ice analyst and the ConvNets, improving their predictive
performance. Preliminary results from a multi-parameter retrieval using a multi-headed version of ASIP allowed to learn to
climatological sea ice conditions support this, showing a significantly improved predictive performance on SoD, but further
exploration is warranted (Wulf et al., 2023b).


Ice charts produced at the national ice services consist of uniform polygons with assigned ice attributes in adherence to the
SIGRID3 format (on Sea Ice SIGRID-3, 2014). This simplified and comparatively coarse representation of the sea ice cover
constrains the utility of ice charts as label data for training SAR-based deep learning models, as it limits their capability to
fully exploit the wealth of information provided in the SAR observations. Furthermore, ice charts lack several important sea
ice features, such as leads, ridges, and melt ponds, which are therefore not learned by the model. These limitations underscore
the need for improved label data that enables deep learning models to better exploit SAR observations. Although other sources



of label data exist, e.g. in-situ observations or labels derived from high-resolution optical imagery from space-borne sensors, they are often sparse in availability, both in terms of the number of data points and geographical and/or all-season coverage. The primary advantage of ice charts remains their abundance and widespread availability, while the challenge of gathering

high-quality label data from other sources persists. A potential avenue to address this issue is the leveraging of self-supervised learning techniques to exploit the vast archive of readily available SAR observations from sources such as the Sentinel-1 mission. In recent years, self-supervised pre-training of vision models (He et al., 2021; Caron et al., 2021) has emerged as a powerful alternative to supervised pre-training using large labeled datasets, e.g. ImageNet. These pre-trained models, occasionally referred to as *Foundation Models* (FMs), can be fine-tuned using smaller labeled datasets for various downstream

applications with competitive performances (He et al., 2021; Caron et al., 2021). Recently, these techniques have entered the domain of Earth Observation, with self-supervisory training strategies being applied to SAR imagery from Sentinel-1 (Allen et al., 2023), to optical imagery from Landsat and Sentinel-2 (Jakubik et al., 2023), and to Sentinel-1 and Sentinel-2 imagery in combination (Fuller et al., 2022). Allen et al. (2023) demonstrate that self-supervised pre-training on Sentinel-1 imagery using masked autoencoding (He et al., 2021) drastically reduces the number of labeled samples required to achieve competitive pre-

dictive performances on various downstream tasks. A SAR-based FM pre-trained using self-supervisory training strategies on the vast archive of Sentinel-1 imagery covering the polar regions has the potential to mitigate the issue of high-quality label data scarcity in the Arctic and Antarctica, enabling a multitude of polar applications with fewer labeled data points, e.g. mapping of sea ice concentration, type, floe size, deformation, ridges, leads, melt ponds, etc. Furthermore, a SAR-based FM pre-trained on observations from multiple SAR sensors might mitigate generalization issues related to sensor-specific characteristics, e.g.

calibration differences, noise patterns, acquisition modes, etc. This would not only fast-track the uptake of SAR observations from future sensors in operational contexts, but also enable applications built on sensors that are currently in operation to be more easily generalized to observations from past SAR missions (e.g. an application built on Sentinel-1 generalized to Envisat ASAR), paving the way for the generation of SAR-based Climate Data Records.

## 6   Conclusions

This work presents ASIP, a new deep learning-based methodology to retrieve high-resolution sea ice concentration and associated uncertainties from SAR and passive microwave observations at a pan-Arctic scale for all seasons. ASIP is an ensemble model consisting of U-Net-like ConvNets trained on the ASIDv2+ dataset, with Sentinel-1 HH/HV imagery and AMSR2 brightness temperatures as input and manually produced ice charts from the Greenland and Canadian Ice Services as label data. The trained ConvNets output pseudo-probabilistic predictions on an $80\ m$ grid, which are calibrated using a learned

post-hoc linear transformation. We propose a new metric to quantify miscalibration and use reliability diagrams to show that the employed recalibration technique significantly improves the calibration of the ensemble output, both in terms of class-wise calibration and calibration across confidence regions. Finally, we propose a novel retrieval methodology to retrieve sea ice concentration and the associated uncertainty from the calibrated ensemble output. An initial quantitative evaluation of ASIP's predictive performance against a test set of manually produced ice charts showed good agreement, achieving an $R^2$-score



of 95% and a class-weighted RMSE of 15.8%. Similar predictive performances were observed for the freezing and melting
seasons, respectively. We investigated ASIP's predictive performance at a pan-Arctic scale in a comparative study using a
well-established and operational PMW-based L3 sea ice product from OSI SAF. Although the sea ice extent was very similar
in both products, the comparison revealed that ASIP consistently produced relatively higher sea ice concentration estimates,
with mean biases for the pan-Artic region ranging from 1.45% to 8.55%, and that the discrepancies were primarily attributed

to disparities in the marginal ice zone, i.e. at the intermediate sea ice concentrations. While these discrepancies might stem
from systematic biases in the ice charts learned by ASIP during the training process, it was shown qualitatively that passive
microwave-based sea ice products on occasion underestimate intermediate sea ice concentrations. The main strength of ASIP
(and SAR-based sea ice retrievals in general) remains its high spatial resolution, while the Arctic coverage is poor compared
to passive microwave-based sea ice products.

Pan-Arctic sea ice products based on the ASIP methodology will be operationally provided as part of the Copernicus Marine
Service product portfolio by the end of 2024.

*Data availability.*  The ASIDv2+ dataset used in this study is an extended version of the AI4Arctic / ASIP Sea Ice Dataset - version 2
(ASIDv2) that is freely available: https://doi.org/10.11583/DTU.13011134.v3 (Saldo et al., 2020). ASIDv2+ will be published at a later
stage.

*Author contributions.*  TWU, JBH, SSI and MBK conceptualized the work. TWU and JBH curated the datasets. TWU designed and imple-
mented the methodology, conducted the data analysis and drafted the manuscript. All authors provided insights regarding the interpretation
of data and reviewed and edited the manuscript.

*Competing interests.*  Authors declare no competing interest.

*Acknowledgements.*  This study was funded partly by DMI, the ESA AI4Arctic project (ESA/Contract No. 4000129762/20/I-NB CCN1) and
by the Copernicus Marine Service (Contract No. 23137L00-COP-SEA ICE OBSERV PRODUCTS-9000).

The authors would like to acknowledge the European Commission and the European Space Agency for the provision of Copernicus
Sentinel-1 data, the Japan Aerospace Exploration Agency for the provision of AMSR2 data, the DMI Greenland Ice Service and the Canadian
Ice Service for making their ice charts available for the sea ice dataset.



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
