# Peer review of "Pan-Arctic Sea Ice Concentration from SAR and Passive Microwave"

_EGUsphere, 2024_

## Author Comment (AC1)

*In the text below, **reviewer comments**, author comments, original manuscript text and updated manuscript text are color-coded as shown here.*

Review of " Pan-Arctic Sea Ice Concentration from SAR and Passive Microwave" by Wulf et al.

Robust all-weather multi-sensor SIC estimates for the Arctic are an important topic in cryospheric Earth Observation. Here, the authors present a deep learning-based retrieval framework for SIC which combines SAR and PMW observations, trained with regional ice charts from Danish and Canadian Ice Services.

The authors mainly do a good job of explaining the technically complex training and retrieval. The idea of applying the full classification probability vector's information to estimate SIC is clever and commendable. Still, some gaps in presentation remain. One result figure was missing from the pdf, and some aspects of the retrieval and training could be better justified.

**Reviewer comment**: Also, the pan-Arctic applicability discussion, while logical and relevant, seemed to focus on resolution aspects and operator-dependent biases. I was missing some critical thought on the broader aspects of deriving SIC across the full width of the Arctic Ocean. Once these issues are remedied, however, I see this paper as a very worthwhile addition to the body of SIC retrieval literature, with clear advances in several aspects and a clean delivery in written form.

Author comment: Thanks for the comment. We added a couple of paragraphs in the Discussions section (Section 5) addressing broader aspects of deriving SIC from SAR across the full width of the Arctic Ocean.

We included a small section on currently poor Arctic coverage of Sentinel-1 imagery and the prospect of future sensor-agnostic SAR-based sea ice retrieval algorithms with greater Arctic coverage, e.g. by including SAR imagery from the RADARSAT Constellation Mission in addition to Sentinel-1.

Added to the "Future Work" section of the manuscript:

**Improving the Arctic coverage of SAR-based sea ice retrievals**

Since the loss of Sentinel-1B in December, 2021, the acquisition of Sentinel-1 imagery in the Arctic has been severely reduced, with no images being acquired in the central Arctic at all. The reduced coverage of Sentinel-1 imagery in regions of high maritime traffic has impacted the national ice services that now are more reliant on other SAR missions to meet user demands on the update frequency of their sea ice products. For example, the Radarsat Constellation Mission from the Canadian Space Agency provides C-band SAR imagery in the Arctic. A natural next step in the development of SAR-based sea ice retrieval algorithms, such as ASIP, is the adaptation to SAR imagery from multiple sensors. The development of sensor-agnostic SAR-based sea ice retrieval algorithms would greatly improve the coverage of the derived sea ice products in the Arctic, which would benefit not only the national ice services, but the sea ice modeling community as well.

We added a few sentences about the representation of Arctic sea ice conditions in the ASIDv2+ training dataset, which only covers a part of the Arctic. This also relates to the reviewer's very last comment about "the applicability of the near-coastal training data to the full range of ice behaviour across the broad swath of the Arctic Ocean".

Original text: ….The ice charts are often produced on the basis of SAR images, enabling the compilation of training datasets consisting of very timely - if not exact - match-ups between ice chart and SAR image, which is important due to the high spatial resolution of the SAR sensor and the continuous movement of drifting sea ice. However, as ice charts are drawn by ice analysts by manual interpretation of satellite observations, there are bound to be inherent uncertainties in the ice charts, such as analyst subjectivity, and inter - and intra-analyst variability…..

Updated text: ….The ice charts are often produced on the basis of SAR images, enabling the compilation of training datasets consisting of very timely - if not exact - match-ups between ice chart and SAR image, which is important due to the high spatial resolution of the SAR sensor and the continuous movement of drifting sea ice. However, while the ice charts in the ASIDv2+ dataset cover diverse sea ice conditions in the Greenland waters as well as the Canadian Arctic, the dataset might not be representative of all possible sea ice conditions across the full width of the Arctic Ocean. As such, there might be sea ice conditions that are not represented in the ASIDv2+ dataset and thus not available for ASIP to learn. Furthermore, as ice charts are drawn by ice analysts by manual interpretation of satellite observations, there are bound to be inherent uncertainties in the ice charts, such as analyst subjectivity, and inter - and intra-analyst variability…..

**Reviewer comment**: The regional DMI and CIS coverages appear to have overlap near west Greenland. Did the authors assess the similarity of the ice charts as a measure for subjective analyst's classification uncertainty? Section 4.2 suggests the results do reflect some operator dependence, but did you try to quantify it?

Author comment: Thanks for the comment – we did not quantify the ice analyst uncertainty in this study. When we compiled the ASIDv2+ training dataset, we did find a smaller number of Sentinel-1 scenes (<100) in the Baffin Bay region for which we were able to find both DMI and CIS regional ice charts that met our match-up criteria. We agree that it would be beneficial to measure analyst uncertainty by assessing the similarity between ice charts drawn by different analysts on the basis of the same Sentinel-1 acquisition, but we did not include it in the (already wide) scope of the present paper. Such studies on the assessment of inter-analyst variation exist, and we refer to these studies in section 4.2.

**Reviewer comment**: Why were the validation and test datasets very, very small compared to the training dataset? Often at least 10% of all data are assigned to validation and test groups, here it's ~1%. How much does this influence the results?

Author comment: Our focus in this study is the pan-Arctic application of a deep learning model trained on a regional dataset covering only the Greenland waters and parts of the Canadian Arctic. As we cannot assess the quality of ASIP in the pan-Arctic region using a regional dataset of ice charts, we consider the Arctic-wide comparison of the ASIP SIC to the OSI SAF SIC product (OSI-408) as the main result in this study and the main evaluation of the presented methodology. The initial quantitative evaluation of the ASIP SIC against ice charts (section 4.2 and Fig. 5) is included in the manuscript to show the result of the training of the ConvNets, as well as the improvement in SIC accuracy (measured against ice charts) over previous studies when using our proposed SIC retrieval from the calibrated confidence vectors. As ASIP is trained with ice charts as labels, the qualitative evaluation in section 4.2 (Fig. 6) is included to show how well (if at all) the model imitates ice charts – both in terms of spatial resolution and quality/accuracy. For these purposes, we would argue that a test set of 50 samples is sufficient, especially given that the test samples have been carefully selected to be representative of the Greenland and Canadian sea ice regimes.

That being said, an 80%/10%/10% split is a good rule of thumb in many cases. However, when the dataset becomes increasingly large, the ratio of test data to training data often diminishes. The ASIDv2+ dataset (5382 samples) is many times larger than the similar datasets ASIDv2 (461 samples) and AI4Arctic Sea Ice Challenge Dataset (533 samples). Other studies using these previous, smaller versions of the ASIDv2+ dataset, e.g. Stokholm et al. (2022) and Chen et al. (2023) use test sets of 23 and 20 scenes, respectively.

**Reviewer comment**: Section 3.2. answers well the question of "what", but offers little for the question "why". Did you test alternative deep learning methods than the ConvNet chosen? What was the key reason to choose it over other alternatives?

Author comment: Thanks for the comment. We did not test other deep learning methods. Our focus in this study is the generalization of a regionally trained model to the pan-Arctic region as well as the introduction of a new method for quantifying the uncertainty of the sea ice products inferred by the trained model - two subjects missing in the current corpus of literature on ML-based sea ice retrievals from SAR. For this reason, we chose a fairly simple UNet model, which is widely used in Earth Observation. We "modernized" the original UNet (which is an "old" architecture now, from 2015) with findings from two widely recognized papers from the field of computer vision, namely Sandler et al. (2018) and Liu et al. (2022). We added a sentence in the beginning of Section 3.2 to provide the reader with the reasoning for choosing the chosen ConvNet architecture.

Original text: The ConvNet we employ in this study follows a modified U-Net (Ronneberger et al., 2015) structure…

Updated text: As the focus of this work is on the generalization of deep learning-based sea ice retrieval algorithms and the uncertainty quantification of their outputs, rather than on the architectural optimization of the predictive performance of the algorithm, we carry out all

experiments in the subsequent sections using a fairly simple ConvNet architecture. The architecture follows a modified U-Net (Ronneberger et al., 2015) structure…

**Reviewer comment**: Figure 9 seemed to be missing entirely from the pdf?

Author comment: We are sorry to hear about this inconvenience. However, when we download the .pdf from the Cryosphere MS records, Fig. 9 is there on page 23, as well as in the following link from the discussion page: https://egusphere.copernicus.org/preprints/2024/egusphere-2024-178/egusphere-2024-178.pdf.

**Reviewer comment**: Fig 10 and similar – the difference plot color range would be better constrained to typical observed ranges rather than the physical maxima of +/- 100%.

Author comment: We agree that the dynamic range should be lower to highlight the spatial variation in the difference plots. While the majority of the values in the difference plots (ASIP SIC – OSI SAF SIC) in Figs. 9 and 10 are positive and between 0% and ~60%, there are both positive and negative extremes in all plots. We believe it's important to include both negative and positive values in the differences plots for the reader to study. We changed the dynamic from the physical maxima of +/- 100% to +/- 50%.

Minimum and maximum values for all difference plots are listed below.

Figure – date: minimum, maximum

Fig. 9 – Jan 24th: -90, 100

Fig. 9 - May 28th: -97, 100

Fig. 9 – Sep 24th: -100, 100

Fig. 10 - Jan 13th: -89, 99

Fig. 10 - April 21[st]: -96, 100

Fig. 10 – Aug 3rd: -87, 100

**Reviewer comment**: While I agree that the pan-Arctic SIC estimates here appear reasonable, I hold some reservations about the applicability of the near-coastal training data to the full range of ice behaviour across the broad swath of the Arctic Ocean. For example, were there sufficient leads and melt ponds in summer in the training data w.r.t. the innermost AO? Were ridges present in the full height range encountered?

Author comment: Thanks for the comment. The applicability of a regionally trained model to the pan-Arctic region is indeed one of the main research questions in this work. The ASIDv2+ dataset, which is used for the training of ASIP, covers sea ice conditions in the Greenland waters as well as the Canadian Arctic. This geographic region is diverse in terms of sea ice

conditions, covering the full seasonal cycle of sea ice freeze-up and melt, and multi-year ice represented in parts of the Canadian Archipelago, north of Greenland and along the eastern coast of Greenland. Still, we cannot be sure that all sea ice behaviors are represented in the training dataset and thus available for the model to learn. In the comparison against OSI-408-a (Figs. 9, 10 and lines 426-447 in the main text), our objective was to utilize OSI SAF SIC as a benchmark for our ASIP SIC product at a pan-Arctic scale. This comparison aimed to demonstrate that our SAR-based sea ice concentration product aligns well with an established and trusted standard. The OSI SAF Sea Ice Concentration (SIC) algorithm is a highly reliable and time-tested tool that has been in operational use for decades. OSI SAF SIC products are integral to renowned operational and climate models.

Regarding the reviewer's point on the representation of leads and melt ponds during summer in the training dataset, we attempt to address some of these issues in the discussion on the limitations of ice charts as label data for training these algorithms (Sec. 5, lines 519-520). While leads and melt ponds are definitely represented in the Sentinel-1 imagery contained in the ASIDv2+, these ice features are not delineated in the corresponding ice charts, and therefore, these features are not explicitly learned by the model. In the future, ice features, such as leads, could be manually added to the ice charts in ASIDv2+ to allow the model to map these features as well.

**References:**

A. Stokholm, T. Wulf, A. Kucik, R. Saldo, J. Buus-Hinkler and S. M. Hvidegaard, "AI4SeaIce: Toward Solving Ambiguous SAR Textures in Convolutional Neural Networks for Automatic Sea Ice Concentration Charting," in *IEEE Transactions on Geoscience and Remote Sensing*, vol. 60, pp. 1-13, 2022, Art no. 4304013, doi: 10.1109/TGRS.2022.3149323.

Chen, X., Patel, M., Pena Cantu, F., Park, J., Noa Turnes, J., Xu, L., Scott, K. A., and Clausi, D. A.: MMSeaIce: Multi-task Mapping of Sea Ice Parameters from AI4Arctic Sea Ice Challenge Dataset, EGUsphere [preprint], https://doi.org/10.5194/egusphere-2023-1297, 2023.

M. Sandler, A. Howard, M. Zhu, A. Zhmoginov and L. -C. Chen, "MobileNetV2: Inverted Residuals and Linear Bottlenecks," *2018 IEEE/CVF Conference on Computer Vision and Pattern Recognition*, Salt Lake City, UT, USA, 2018, pp. 4510-4520, doi: 10.1109/CVPR.2018.00474.

Liu, Z., Mao, H., Wu, C., Feichtenhofer, C., Darrell, T., and Xie, S.: A ConvNet for the 2020s, CoRR, abs/2201.03545, https://arxiv.org/abs/2201.03545, 2022

---

## Author Comment (AC2)

*In the text below, **reviewer comments**, author comments, original manuscript text and updated manuscript text are color-coded as shown here.*

Anonymous Referee #1, 27 Mar 2024

Review for

Pan-Arctic Sea Ice Concentration from SAR and Passive Microwave

by

Tore Wulf, Jørgen Buus-Hinkler, Suman Singha, Hoyeon Shi, and Matilde Brandt Kreiner

General comments:

Considering the need for wide-coverage regular monitoring of environmental and climatological changes in the Arctic, this paper provides an interesting contribution, dealing with automated retrieval of sea ice concentration (SIC) at higher spatial resolution. The key points are: (1) the introduction of a deep learning based SIC retrieval with improved spatial resolution and with associated calibrated uncertainties, and (2) the use of a substantially extended training and validation data set of Sentinel-1 images collocated with AMSR-2 data which covers the whole periphery of Greenland, the Canadian Archipelago, and parts of the Labrador Sea, and contains more than 5000 samples acquired from 2018 to 2021.

The retrieval method is based on an ensemble of convolutional neural networks (ConvNets) and includes investigations of re-calibration strategies and metrics for quantifying mis-calibration. With a proper calibration of the retrieval method one guarantees that the confidence scores provided by the model reflect its predictive uncertainty which is needed by the end-user to directly assess the reliability of the SIC information (that is how I understood it).

The paper is well structured, and the text well formulated, although in parts at a too detailed level without first providing the main questions. In my opinion the paper should definitely be accepted with considerations of the suggestions and comments provided below.

I have two main issues

**Reviewer comment**: (1) Methodology and Future Work :

Subsections 3.3.1 to 3.3.3, and Sect. 3.4 present many details. I had some difficulties to follow the use of single recalibration strategies and single mis-calibration metrics. Apparently all of the former are combined with all of the latter? Which then helps to decide which

ConvNets configuration is finally used (lines 381-383 in Sect. 4.1)? I recommend to provide a graphical presentation of the workflow that is described in the sections mentioned above. This will help the reader to understand the overall structure of the methodology before digging into the many details provided in the recent text. In addition it would be helpful if the authors formulate the motivating questions, which in my understanding are roughly: Which are the optimal recalibration strategies? Which are the optimal metrics to decide?

Author comment: Thanks for the comment. As the reviewer notes, many details are presented in Section 3.3 and 3.4, and it can be difficult to follow as the motivation for each step along the way might not be evident to the reader. Rather than a graphical presentation - as the reviewer suggests -, we included an overview of the contents and workflow in the Methodology section to help guide the reader.

Added to the manuscript: This section is organized as follows.

- In section 3.1 we present the details of how the ASIDv2+ dataset is prepared for the training, calibration and initial evaluation of our proposed SIC retrieval.
- In section 3.2 we present the architecture of the ConvNet employed in this study.
- In section 3.3 we delve into the concept of calibration - or inversely, *miscalibration* - in the context of deep learning-based classifiers. We present a widely used metric that quantifies miscalibration, identify its shortcomings, and introduce a new metric that addresses the identified shortcomings. We also introduce reliability diagrams as a way of qualitatively assessing the calibration of a classifier. Then we present multiple recalibration strategies that have been proposed to rectify miscalibration. Lastly, we propose a novel SIC retrieval from a *well-calibrated* classifier output that retrieves a continuous SIC field as well as the associated uncertainty field characterized by a standard deviation.
- In section 3.4 we present the details of the experimental setup of the study, including the details of the optimization strategy and training of the ConvNet. We set up experiments to evaluate the effectiveness of the presented recalibration strategies in their ability to reduce miscalibration using the miscalibration metrics and reliability diagrams as performance measures. Having determined the most effective recalibration strategy, we set up experiments to evaluate the predictive performance of our proposed SIC retrieval against regional ice charts and at a pan-Arctic scale against the well-established OSI-408-a SIC product from OSI SAF.

Author comment: To the same effect of providing the reader with a clearer motivation of the work, we changed the wording of the last paragraph of the introduction to emphasize the overall subjects of the work, namely the generalization of a regionally trained model to the pan-Arctic region as well as the introduction of a new method for quantifying the uncertainty of the sea ice products inferred by the trained model.

Original text: In this paper, we present a new and comprehensive deep learning-based SIC retrieval methodology denoted *ASIP* (Automated Sea Ice Products). ASIP is an ensemble of ConvNets retrieving high-resolution SIC with accompanying well-calibrated uncertainties from Sentinel-1 SAR imagery and AMSR2 brightness temperatures. ASIP is trained on a new, vast

training dataset with Sentinel-1 HH/HV imagery and Advanced Microwave Scanning Radiometer 2 (AMSR2) brightness temperatures as input and manually produced ice charts from the Greenland and Canadian Ice Services (CIS) as labels. We explore several recalibration strategies and introduce a new metric to quantify miscalibration for imbalanced multi-class classification tasks. Using reliability diagrams, we show that our proposed metric surpasses the popular ECE (Expected Calibration Error) metric, particularly when it comes to identifying class-wise miscalibration and miscalibration across confidence regions. We propose a new retrieval methodology to retrieve SIC and the associated uncertainty from the calibrated ensemble output. Finally, we show that ASIP generalizes well to the pan-Arctic region in all seasons in a comparative study against a well-established and operational PMW-based SIC product.

Updated text: In this work we address two subjects that are missing in the current corpus of ML-based sea ice retrievals from SAR, namely the generalization of regionally trained sea ice retrieval algorithms to the pan-Arctic region and the uncertainty quantification of the sea ice products inferred by such algorithms. We present a new and comprehensive deep learning-based SIC retrieval methodology denoted *ASIP* (Automated Sea Ice Products), capable of retrieving high-resolution SIC estimates with accompanying well-calibrated uncertainties from Sentinel-1 SAR imagery and brightness temperatures from the Advanced Microwave Scanning Radiometer 2 (AMSR2). ASIP is trained on a new, vast training dataset with Sentinel-1 HH/HV imagery and AMSR2 brightness temperatures as input and manually produced ice charts from the Greenland and Canadian Ice Services (CIS) as labels. In a comparative study using a well-established and operational PMW-based SIC product as a baseline, we show that ASIP generalizes well to the pan-Arctic region for all seasons.

**Reviewer comment**: In the sub-section "Future work" it would be helpful for the reader when - with one or two sentences - the overall topic of this part is introduced, which in my understanding is the discussion of two alternative methods: multi-parameter retrieval (lines 489 to 525) and self-supervised learning (starting line 525), and separate these two alternatives also visually in the text formatting.

Author comment: Thanks for the comment. In the Future work subsection we discuss two ideas for further development of algorithms for SAR-based sea ice retrieval. These ideas are not necessarily alternatives to the proposed methodology of the manuscript, but rather add-ons or avenues for further research/development. We have separated the two parts visually in the manuscript formatting and provided a headline for each part of the Future Work subsection, "Multi-parameter sea ice retrieval" and "Limitations of the use of ice charts as label data and self-supervised learning as a way forward".

**Reviewer comment**: (2) Figs 8-12 and corresponding text (lines 395 -447).

With the given figures, judgements regarding the similarity between the ASIP results and the ice charts shown for comparison are possible only on a subjective basis. Even the authors

themselves use only vague formulations: "resemble …to a significant extent (lines 398-399) and "fairly similar" (line 440). In my (subjective) opinion, the differences between ASIP results and the data shown for comparison are relatively large in some regions. In particular the results shown on the pan-Arctic scale are difficult to assess, even with zooming (here in particular Fig. 7 and 8, since the interpretation of the SAR images in terms of SIC will be difficult for most readers). I recommend to keep Fig. 7 as an example for the incomplete Sentinel-1 coverage in comparison to the final ASIP SIC results, and replace Fig. 8 with regional (scale of a Sentinel-1 EW scene) examples that show cases of higher and lower uncertainties, with possible explanations in text for the latter. Figure 9 and 10 could be combined into a single figure, choosing the three most interesting cases including the bottom row of Fig 10 (H-L) for which zoom-ins are shown in Figs. 11-12. I would also like to see the ASIP uncertainty maps in Figs. 9-12 which are more important for a judgement of possible problems than the difference to the OSI-SAF SIC. Because of the uncertainties in the OSI-SAF data the difference maps do not help to judge the accuracy of the ASIP. If possible and if they want, the authors could provide more figures as supplementary material.

Author comment: Thanks a lot for the comments. The reviewer raises several concerns about Figures 8-12 and the corresponding text (lines 395-447). For clarification, the manuscript lines 395-421 refer to Figure 6, while lines 422-447 refers to Figures 7-12. We separate the reviewer comments above into two parts to answer the comments separately:

**Reviewer comment**: With the given figures, judgements regarding the similarity between the ASIP results and the ice charts shown for comparison are possible only on a subjective basis. Even the authors themselves use only vague formulations: "resemble …to a significant extent (lines 398-399) and "fairly similar" (line 440).

Author comment: Lines 398-399 refer to Figure 6, which visually compares the ASIP SIC output to a subset of regional ice charts from the ASIDv2+ test set. This figure allows the reader to qualitatively assess the quality and resolution of the ASIP SIC output, as well as the accompanying uncertainty. A quantitative assessment of the similarity (or the predictive performance) of ASIP against the full ASIDv2+ test set is provided in the main text of the same section (4.2), with R2 and RMSE as summarizing statistics, and Figure 5 showing the spread across the full range of sea ice concentrations. Indeed, judgements regarding the visual similarity between ASIP and the regional ice charts in Figure 6 will be subjective. We removed the line with the subjective assessment of the similarity between ASIP and the regional ice charts in Figure 6.

Original text: All examples are depicted in the original Sentinel-1 SAR geometry and the geographical extent of each example is outlined in Figure 1. Generally, the ASIP retrieval produces sea ice maps that resemble the manually produced ice charts to a significant extent. However, as the manually produced ice charts are drawn as smooth delineated polygons of relatively homogeneous sea ice conditions, the sea ice maps produced using the ASIP retrieval might contain more detail and variability, with a larger degree of similarity to the spatial patterns and textural intricacies in underlying SAR imagery.

Updated text: All examples are depicted in the original Sentinel-1 SAR geometry and the geographical extent of each example is outlined in Figure 1. As the manually produced ice charts are drawn as smooth delineated polygons of relatively homogeneous sea ice conditions, the sea ice maps produced using the ASIP retrieval might contain more detail and variability, with a larger degree of similarity to the spatial patterns and textural intricacies in underlying SAR imagery.

Author comment: Line 440 refers to Figure 11, which shows a zoomed-in view of Figure 10I-L from the Kara Sea, highlighting discrepancies between the ASIP SIC and the OSI SAF SIC. This particular line mentions the similarity in the sea ice extents in the ASIP SIC and the OSI SAF SIC, and it does not mention ice charts. We removed the subjectivity from the line.

Original text: A zoomed-in view spanning the region from Franz Josef Land toward the Kara Sea with the accompanying Sentinel-1 imagery is shown in Figure 11. While the extent of the sea ice is fairly similar in ASIP and OSI SAF, the sea ice concentrations are significantly higher in the ASIP retrieval.

Updated text: A zoomed-in view spanning the region from Franz Josef Land toward the Kara Sea with the accompanying Sentinel-1 imagery is shown in Figure 11, exemplifying the sea ice concentration discrepancies between ASIP and OSI SAF.

**Reviewer comment**: In my (subjective) opinion, the differences between ASIP results and the data shown for comparison are relatively large in some regions. In particular the results shown on the pan-Arctic scale are difficult to assess, even with zooming (here in particular Fig. 7 and 8, since the interpretation of the SAR images in terms of SIC will be difficult for most readers). I recommend to keep Fig. 7 as an example for the incomplete Sentinel-1 coverage in comparison to the final ASIP SIC results, and replace Fig. 8 with regional (scale of a Sentinel-1 EW scene) examples that show cases of higher and lower uncertainties, with possible explanations in text for the latter. Figure 9 and 10 could be combined into a single figure, choosing the three most interesting cases including the bottom row of Fig 10 (H-L) for which zoom-ins are shown in Figs. 11-12. I would also like to see the ASIP uncertainty maps in Figs. 9-12 which are more important for a judgement of possible problems than the difference to the OSI-SAF SIC. Because of the uncertainties in the OSI-SAF data the difference maps do not help to judge the accuracy of the ASIP. If possible and if they want, the authors could provide more figures as supplementary material.

Author comment: Our focus in this study is the pan-Arctic application of a deep learning model trained on a regional dataset covering only the Greenland waters and parts of the Canadian Arctic. It is therefore essential to show that the learned model generalizes well beyond the geographical and temporal bounds of the training dataset. The Pan-Arctic results in section 4 show the geographic generalization (as the training dataset covers Greenland water and Canadian Arctic), and the 2023 results show the temporal generalization (since the training dataset spans 2018-2021). Hence, we argue that Figures 7-8 (which show Sentinel-1 HH, ASIP SIC and ASIP Uncertainty mosaics) and Figures 9-10 (which show the pan-Arctic comparison) that show pan-Arctic results for the freezing season, melting season and around the yearly minimum for 2020 and 2023, respectively, are the main results from this study and

should be included in the manuscript in their full form. We agree with the reviewer that it is difficult to assess the results at a pan-Arctic scale. We opted to not combine Figures 7-10 into two figures (one for 2020 and one for 2023), as this would clutter the figures further, and make it more difficult for readers to assess the results and zoom in.

The reviewer suggests including a figure with ASIP SIC results at a regional scale (or scale of a Sentinel-1 EW scene). We argue that such examples are already present in the manuscript. Figure 6 shows 5 examples of single Sentinel-1 EW scenes with corresponding ASIP SIC/Uncertainty results and regional ice charts, with possible explanations for varying uncertainties given in the main text. In Figure 10I-L, from August 2023, we observe the largest discrepancies between ASIP and OSI SAF, and therefore, we select two zoomed-in regional views from these mosaics to highlight cases in which the differences between ASIP and OSI SAF are particularly pronounced (Figures 11-12) - both in terms of sea ice concentration and spatial resolution of the respective products.

As suggested by the reviewer, we have added the ASIP uncertainty maps to Figures 11 and 12 and changed the figure captions accordingly. The ASIP uncertainty maps for Figures 9 and 10 are available in Figures 7 and 8. We added a sentence in the captions for Figures 9 and 10 to refer the reader to Figures 7 and 8 for the corresponding Sentinel-1 imagery and ASIP uncertainty maps.

Manuscript text: Pan-Arctic comparisons between ASIP and OSI SAF for three 7-day periods in 2020. From top to bottom, 7-day periods are; Jan 9th to Jan 15th, April 17th to April 23rd, and July 31st to Aug 6th. NIC ice charts are included as an additional reference. From left to right; NIC ice chart, OSI SAF SIC mosaic, ASIP SIC mosaic, difference map between ASIP SIC and OSI SAF SIC. The corresponding Sentinel-1 HH mosaics and ASIP uncertainty maps can be seen in Figure 7.

While it is true, as noted by reviewers, that OSI SAF products have inherent uncertainties, particularly in the Marginal Ice Zone, our objective was to utilize OSI SAF SIC as a benchmark for our ASIP SIC product. This comparison aims to demonstrate that our SAR-based sea ice concentration product aligns well with an established and trusted standard. The OSI SAF Sea Ice Concentration (SIC) algorithm is a highly reliable and time-tested tool that has been in operational use for decades. OSI SAF SIC products are integral to renowned operational and climate models.

Minor questions:

(some of my questions are related to my lack of knowledge of deep learning terminology)

**Reviewer comment**: lines 176-177: Are single NIC charts composed of observational data acquired at different days, or of data from just one fixed day which may be from up to 5 days prior to production of the ice chart?

Author comment: The NIC charts are based on observational data from up to five days prior to the date of the chart. These observational data can be acquired on different days. We

made a small change to the manuscript to emphasize this distinction. We also added a reference to the NIC chart user guide that contains this (and more) information.

Original text: The ice charts are based on observational data acquired up to five days prior to the issue date of the ice chart, and thus, the ice charts represent the sea ice conditions up to five days prior to the ice chart timestamp.

Updated text: The ice charts are based on observational data acquired during a time period up to five days prior to the issue date of the ice chart, and thus, the ice charts represent the sea ice conditions up to five days prior to the ice chart timestamp.

**Reviewer comment**: lines 198-199: "The Sentinel-1 HH/HV bands and the AMSR2 brightness temperatures are standardized prior to training" - what means standardize?

Author comment: We added further explanation to lines 198-199:

Original text: The Sentinel-1 HH/HV bands and the AMSR2 brightness temperatures are standardized prior to training.

Updated text: The Sentinel-1 HH/HV bands and the AMSR2 brightness temperatures are standardized prior to training by subtracting the mean and scaling to unit standard deviation. The means and standard deviations of the Sentinel-1 backscatter intensities and the AMSR2 brightness temperatures are computed from the ASIDv2+ training set.

**Reviewer comment**: Fig. 3: what is the meaning of HxWxC, HxWx(RC)? Although the other abbreviations are explained in the text, I recommend to repeat the explanations in the figure caption which makes it easier for the reader to understand the graph without jumping back and forth between text and figure. For readers like me who are not familiar with deep learning terminology, it is helpful to explain (or replace) the "Conv1x1" which probably means to map an input pixel to an output pixel without considering the pixels around (so in fact there is no convolution).

Author comment: We added a more detailed figure caption explaining the structure of the residual block in Figure 3.

To your second point: a pointwise convolution (or Conv1x1) is a type of convolution that uses a 1x1 kernel with the specific purpose of performing a pixel-wise mapping from one feature space to another (e.g. for dimensionality reduction). While the special case of convolutions with 1x1 kernels can be expressed in more simple terms, the Conv1x1 terminology is used extensively within the field of computer vision, e.g. in the context of depthwise-separable convolutions, as is the case in this paper. The implementation of 1x1 convolutions is also very different from fully connected layers and these are not directly interchangeable. We would argue that our application of pointwise convolutions is explained in section 3.2.

Original text: Structure of the inverted residual block used in the ConvNet.

Updated text: Structure of the inverted residual block used in the ConvNet. The block consists of a pointwise convolution (Conv1x1) that projects the input feature maps of size HxWxC (Height x Width x Channels) from a C-dimensional feature space to an RC-dimensional feature space with an expansion factor R, a depthwise convolution (DWConv3x3) followed by Batch Normalization (BN) and the Gaussion Error Linear Unit (GELU), and lastly, a pointwise convolution that reprojects the feature maps from an RC-dimensional feature space back to a C-dimensional feature space followed by a LayerScale (LS) operation.

**Reviewer comment**: Equation (1): What is parameter "y" in words?

Author comment: "y" is the class. In equation 1, p_i(y=i|x) reads as the probability of the class y being i given observations x. We've updated the manuscript so the meaning of "y" is explicitly stated in the text.

Original text: In the following, we consider a classifier with $k$ classes 1, …, k.

Updated text: In the following, we consider a classifier with $k$ classes $y \in \{ y_1, y_2, \ldots, y_k \}$.

**Reviewer comment**: Equation (2): How is the accuracy determined? What means "support of bin $B_m$"?

Author comment: The accuracy of bin $m$ is the proportion of the correctly classified samples in bin $m$, i.e. the number of correctly classified samples in bin $m$ divided by the total number samples in bin $m$.

The support of bin $m$ is equal to the total number of samples in bin $m$. We changed it throughout the manuscript such that "support of B_m" is changed to "number of samples in B_m", or similar.

**Reviewer comment**: line 276: what means "hold-out" validation?

Author comment: A hold-out validation dataset is a dataset that is withheld from training and used for evaluation/calibration/hyperparameter tuning, etc. We rephrased to emphasize that the hold-out validation dataset is a dataset that has been separated from the training dataset.

Original text: These scaling approaches use a hold-out validation set to learn a single parameter, or a set of parameters, to rescale the logit vector z before passing z through the softmax function.

Updated text: These scaling approaches use a hold-out validation dataset that has been split from the training dataset to learn a single parameter, or a set of parameters, to rescale the logit vector z before passing z through the softmax function.

**Reviewer comment**: line 288: what is a "one-hot" encoded label?

Author comment: One-hot encoding (also called 'dummy encoding' in statistics) is the transformation of a categorical variable into a set of binary variables. It is a widely used technique in machine learning to represent categorical data using 0's and 1's. For example, if a model is trained to predict the colors red, green and blue, these colors can be represented using one-hot encoding, e.g. red: [1, 0, 0], green: [0, 1, 0], blue: [0, 0, 1]. We added a short bracketed explanation in the manuscript.

Original text: With label smoothing, the target becomes a mixture of the one-hot encoded label and a uniform distribution with a smoothing factor…

Updated text: With label smoothing, the target becomes a mixture of the one-hot encoded label (the categorical label transformed into a set of binary labels) and a uniform distribution with a smoothing factor…

**Reviewer comment**: line 330: …we set "the" bin support…

Author comment: fixed, changed "to" to "the".

**Reviewer comment**: line 350: I do not see much sense in considering NIC charts with a time difference of up to 12 days relative to the actual observations. They can definitely not be used for judging the quality of the ASIP results. But this is not a critical point. The authors could better explain why they included this comparison.

Author comment: We agree with the reviewer that the NIC charts cannot directly be used to judge the quality of ASIP. Both ASIP and OSI SAF sea ice concentrations are derived from satellite observations and both are susceptible to errors. While the NIC charts are by no means the ideal additional reference, we do believe they add some value to the study, especially in cases where the discrepancies between ASIP and OSI SAF are relatively large. One such example is Figure 12, where OSI SAF and ASIP disagree on the presence of sea ice in the Baffin Bay region. We added two sentences to this paragraph to stress that the NIC charts cannot be used to assess the quality of ASIP directly.

Manuscript text: As an additional reference, we show pan-Arctic ice charts produced by NIC with an issue date within the 7-day period. Note, however, that there can be a lag of several days (up to 12 days in the worst case) between the acquisition time of the observations used to generate the mosaics and the acquisition time of the observations used to produce the NIC ice chart. Therefore, the NIC charts cannot be used directly to assess the quality of neither our SIC retrievals nor OSI SAF. Instead, the NIC charts are used as an additional reference in case of large discrepancies between our SIC retrievals and those of the OSI SAF product.

**Reviewer comment**: lines 369-370: "…introduced the during…"?

Author comment: fixed, removed "the".

**Reviewer comment**: line 388: please give a range for "intermediate SIC"

Author comment: We added a range of 20%-80% for the intermediate SIC in the manuscript. This range roughly aligns with results of Cheng et al. 2020 (figure 16).

Original text: The ASIP retrieval achieves an overall $R^2$-score of 95%, with the largest deviations occurring at the intermediate sea ice concentrations.

Updated text: The ASIP retrieval achieves an overall $R^2$-score of 95%, with the largest deviations occurring at the intermediate sea ice concentrations (20%-80%).

**Reviewer comment**: Figs. 6-10: The identification scheme of the single plates (A,B,C…) in the figures, as used in the text, should be explained in at least the caption for Fig. 6, the other figures may refer to it.

Author comment: We added a sentence in the caption for Fig. 6. that explains that each plot in the figure is given a letter identifier.

Original text: 5 examples scenes from the ASIDv2+ test set. From left to right: Sentinel-1 HH, manually produced regional ice chart from CIS or DMI, sea ice concentration retrieved by ASIP, uncertainty reported by ASIP. Zoom in to view details.

Updated text: 5 example scenes from the ASIDv2+ test set. From left to right: Sentinel-1 HH, manually produced regional ice chart from CIS or DMI, sea ice concentration retrieved by ASIP, uncertainty reported by ASIP. Zoom in to view details. Each plot in the figure is given an identifier (a letter, ordered alphabetically) that can be referred to in the text.

**Reviewer comment**: lines 412-414: The increase of the ocean backscatter due to wind does not change the absolute level of backscattering from the sea ice (or did I misunderstand this sentence?). What is changing is the intensity contrast between ice and water which probably is not easy to consider in the training of the ConvNets without additional information about wind conditions.

Author comment: Indeed – the roughening of the ocean surface results in a SAR image in which the backscatter intensities over sea ice are relatively low compared to the backscatter intensities over the open ocean.  We changed the wording a bit to emphasize that it is the intensity *contrast* that is changing, rather than the absolute backscatter intensities from the sea ice.

Original text: In Figure 6M-P the wind-roughening of the ocean surface leads to very high backscatter intensities over open water, particularly in the near- to mid-range, which

consequently leads to the sea ice in the mid - to far-range exhibiting relatively low backscatter intensities in the resulting SAR image.

Updated text: In Figure 6M-P the wind-roughening of the ocean surface leads to very high backscatter intensities over open water, particularly in the near- to mid-range, which consequently leads to a change in the intensity contrast with the sea ice in the mid- to far-range appearing dark in the resulting SAR image.

**Reviewer comment**: Figs. 9 and 10: ASIP SIC - OSI SAF SIC: The range for showing the difference values should be selected smaller, e.g. excluding the negative differences which don't occur in the maps (instead a corresponding hint in the figure caption?)

Author comment: While the majority of the values in the difference plots (ASIP SIC – OSI SAF SIC) in Figs. 9 and 10 are positive and between 0% and ~60%, there are both positive and negative extremes in all plots. We believe it's important to include both negative and positive values in the differences plots for the reader to study. We agree that the dynamic range should be lower to highlight the spatial variation in the difference plots. We changed the dynamic from the physical maxima of +/- 100% to +/- 50%.

Minimum and maximum values for all difference plots are listed below.

Figure – date: minimum, maximum

Fig. 9 – Jan 24th: -90, 100

Fig. 9 - May 28th: -97, 100

Fig. 9 – Sep 24th: -100, 100

Fig. 10 - Jan 13th: -89, 99

Fig. 10 - April 21$^{st}$: -96, 100

Fig. 10 – Aug 3rd: -87, 100

**Reviewer comment**: lines 509-512: "Allowing the ConvNets to learn the location-dependent seasonal variation in sea ice conditions, either by including the location and the time of the year as additional input features or by some other mechanism, we can level the playing field between the ice analyst and the ConvNets, improving their predictive performance." To focus deep learning methods on typical local conditions (either just for retrieval of SIC as stand-alone or of muitl-parameter sets) seems to be the way forward also for improving the accuracy of a pan-Arctic product? Could be explicitly mentioned in the discussion if the authors agree.

Author comment: Thanks a lot for the comment. We agree that allowing the ConvNets to learn the climatological sea ice conditions (that varies with geographic location and time of the year) has potential to improve the accuracy for Arctic-wide products as well. The mechanism by which we introduce this information in the training of the ConvNet can limit the applicability

of the learned model in regions outside of the geographical bounds of the training dataset. For example, a way of allowing the ConvNets to learn the typical local sea ice conditions is to include geographic location and time of the year as additional input features to the ConvNets. This approach, however, introduces a generalization problem if the ConvNets are applied outside the geographical bounds of the training dataset (which only covers the Canadian Arctic and the Greenland Waters). The location and time of the year input features are intrinsically linked to the local sea ice conditions in the training dataset and provides a good learning signal for the ConvNets, but the learned models will be trained specially for the region covered by the training dataset. The coordinates of a new location (e.g. parameterized by X and Y polar stereographic coordinates), for example in the Kara Sea, will have ranges outside of the X, Y coordinates of the samples in the training dataset. Instead, we would need to find a mechanism by which we can introduce information about the climatological sea ice conditions that is also generalisable to regions beyond the bounds of the training dataset OR produce a training dataset that covers and is representative of the entire region (and seasons) of interest.

We added parts of these reflections to the Future work section about multi-parameter sea ice retrieval:

Added to manuscript: Note, however, that the mechanism by which we introduce climatological information in the training of the ConvNet can limit the applicability of the trained model in regions outside of the geographical bounds of the training dataset. For example, if we allow the ConvNets to learn the typical local sea ice conditions by including geographic location and time of the year as additional input features to the ConvNets, then the coordinates of a new location (e.g. parameterized by X and Y polar stereographic coordinates), for example in the Kara Sea, will have ranges outside of the X and Y coordinate distributions learned by the model. Instead, one would have to find a mechanism by which we can introduce information about the climatological sea ice conditions that is also generalisable to regions beyond the bounds of the training dataset, or produce a training dataset that covers and is representative of the entire region (and seasons) of interest.

**Reviewer comment**: References: for the first one (Allen et al. 2023) the journal is missing. Note I did not check all references.

Author comment: Thanks! The work by Allen et al. was presented at the NeurIPS 2023 Workshop on Tackling Climate Change with Machine Learning. We have updated the reference and added a URL as well.